# Primphormer: Efficient Graph Transformers with Primal Representations

Mingzhen He [1]   Ruikai Yang [1]   Hanling Tian [1]   Youmei Qiu [1]   Xiaolin Huang [1]

## Abstract

Graph Transformers (GTs) have emerged as a promising approach for graph representation learning. Despite their successes, the quadratic complexity of GTs limits scalability on large graphs due to their pair-wise computations. To fundamentally reduce the computational burden of GTs, we propose a primal-dual framework that interprets the self-attention mechanism on graphs as a dual representation. Based on this framework, we develop Primphormer, an efficient GT that leverages a primal representation with linear complexity. Theoretical analysis reveals that Primphormer serves as a universal approximator for functions on both sequences and graphs, while also retaining its expressive power for distinguishing non-isomorphic graphs. Extensive experiments on various graph benchmarks demonstrate that Primphormer achieves competitive empirical results while maintaining a more user-friendly memory and computational costs.

## 1. Introduction

Graph representation learning has been successfully applied in various fields, including social network analysis (Li et al., 2023), traffic prediction (Dong et al., 2023), drug discovery (Liu et al., 2023), and more. Much of the research in graph representation learning has focused on Message Passing Neural Networks (MPNNs) which rely on *local* message-passing mechanisms. Although MPNNs have emerged as a powerful approach to short-range tasks that require information exchange among nodes in neighborhoods, MPNNs face inherent limitations such as over-smoothing (Nguyen et al., 2023), over-squashing (Giraldo et al., 2023) in long-range tasks (Dwivedi et al., 2022b).

To overcome the limitations, Graph Transformers (GTs)

which allow each node to *globally* attend to all other nodes is proposed to enable the learning of long-range dependencies within the graph (Rampasek et al., 2022; Chen et al., 2022). While GTs are a promising approach, they suffer from a significant drawback: their quadratic complexity caused by pair-wise computations in self-attention mechanisms, which limits their practical applicability.

The key to reducing the quadratic complexity is to use computationally efficient attention mechanisms. Linear attentions like Performer (Choromanski et al., 2021) and Big-Bird (Zaheer et al., 2020) have been integrated into GTs. However, they need to introduce additional computational overhead, which becomes the dominating source of computation for medium-sized graphs (Rampasek et al., 2022). An alternative approach is sparse attention. Shirzad et al. (2023) introduced Exphormer whose efficiency benefits from the sparsity of graphs. However, the complexity increases to quadratic with the number of nodes as graphs become denser, thereby still limiting its scalability.

To fundamentally enhance the scalability of GTs, it is crucial to avoid pair-wise computations, prompting us to consider the primal-dual relationship in kernel machines. Examples of models leveraging this relationship include the support vector machine (Cortes & Vapnik, 1995), the least squares support vector machine (Suykens & Vandewalle, 1999), and the kernel principal component analysis (Mika et al., 1999). The primal-dual relationship represents pairwise and symmetric similarity in duality as an inner product of feature mappings in the primal space. By solving optimization problems in the primal space with these feature mappings, quadratic complexity can be avoided.

When constructing the primal representation of the self-attention mechanism, we encounter an essential problem that attention scores are inherently asymmetric, violating Mercer's condition (Mercer, 1909), which causes the classical primal-dual discussion to fail. Recent research on primal-dual relationships has sought to explore methods for accommodating asymmetry in kernel machines (Suykens, 2016; He et al., 2023a; Chen et al., 2023). Chen et al. (2023) interpreted the primal-dual relationship of the self-attention on *sequences* through asymmetric kernel singular value decomposition. This approach collects data information through uniformly sampling the sequence under an *in-*

---

[1]Institute of Image Processing and Pattern Recognition, Shanghai Jiao Tong University, Shanghai, China. Correspondence to: Xiaolin Huang <xiaolinhuang@sjtu.edu.cn>.

*Proceedings of the 42$^{nd}$ International Conference on Machine Learning*, Vancouver, Canada. PMLR 267, 2025. Copyright 2025 by the author(s).

*ductive bias* assumption that sequences are ordered. However, this assumption does not hold for graphs, as the structure of a graph is defined by its edges, and the arrangement or ordering of nodes is not explicitly specified, leaving a question about discussing the primal-dual relationship of the self-attention on *graphs*.

**Our contributions.** We propose a novel primal representation for GTs, named *Primphormer*. This method supports asymmetry in self-attention on graphs by introducing an asymmetric kernel trick. It avoids costly pair-wise computations and storage overhead without introducing additional heavy computational burden. The primal-dual analysis reveals that Primphormer can leverage graph information to adaptively adjust the output basis, thereby potentially enhancing the model's flexibility. Since Primphormer is a new architecture for GTs, we are also interested in its theoretical properties. To explore this, we demonstrate that Primphormer serves as a universal approximator for arbitrary continuous functions on a compact domain and that it preserves expressive power in terms of distinguishing non-isomorphic graphs. Through extensive experimental evaluations on various graph benchmarks, we show that Primphormer achieves competitive results while maintaining more user-friendly memory and computational costs.

## 2. Methods

**Notations.** We consider a labeled graph $G = (V, E, \ell)$ with the node and edge sets $V, E$ and the labeling function $\ell$. $|V| = N$, $|E| = M$ denote the numbers of nodes and edges. $[N] := \{1, \cdots, N\}$. $\mathbf{A}(G) \in \{0, 1\}^{N \times N}$ denotes the adjacency matrix where $\mathbf{A}_{u,v} = 1$ iff $\{u, v\} \in E$. We take $b, \boldsymbol{b}, \boldsymbol{B}$ to be a scalar, a vector, and a matrix. The inner product of two vectors is written as $\langle \cdot, \cdot \rangle$. The infinite norm of functions is written as $\| \cdot \|_\infty$. $\mathbb{R}$ denotes the set of real numbers. $\mathbb{R}_+$ denotes the set of real and positive numbers. $\mathrm{Tr}(\boldsymbol{S})$ denotes the trace of a square matrix $\boldsymbol{S}$. $\mathrm{vec}(\boldsymbol{B})$ denotes the vectorization of the matrix $\boldsymbol{B}$, formed by stacking the columns of $\boldsymbol{B}$ into a single column vector. $\otimes$ denotes the Kronecker product. $\mathbf{1}$ and $\mathbf{0}$ denote vectors with all 1 and 0, respectively. Denote $[a, b, c]$ as column concatenation and $[a; b; c]$ as row concatenation. $\boldsymbol{X} := [\boldsymbol{x}_1, \cdots, \boldsymbol{x}_N] \in \mathbb{R}^{d \times N}$ is the embedding matrix for nodes where $\boldsymbol{x}_i \in \mathbb{R}^d$ is the embedding of the $i$-th node. For two graphs $G$ and $G'$, a graph isomorphism is a bijection $\varphi(\cdot) : V_G \to V_{G'}$ such that $\{i, j\} \in E_G$ iff $\{\varphi(i), \varphi(j)\} \in E_{G'}$. Two graphs $G$ and $G'$ are isomorphic if there is a graph isomorphism $\varphi(\cdot) : V_G \to V_{G'}$.

### 2.1. Attention mechanism on graphs

An attention mechanism on a graph $G$ treats nodes $V$ as tokens and is modeled by a fully connected, directed graph with the positional encoding (PE) that encodes the geome-

try of $G$. Its directed edges denote a directed interaction or similarity between two nodes $i, j$, computed by the inner product in the attention mechanism. Mathematically, we define the attention mechanism ATTN as follows,

$$
\begin{cases}
\kappa(\boldsymbol{x}_i, \boldsymbol{x}_j) = \sigma\left(\langle \boldsymbol{q}(\boldsymbol{x}_i), \boldsymbol{k}(\boldsymbol{x}_j) \rangle\right) \\
\boldsymbol{o}_i = \sum_{j=1}^{N} \boldsymbol{v}(\boldsymbol{x}_j) \kappa(\boldsymbol{x}_i, \boldsymbol{x}_j), \quad i, j \in [N],
\end{cases}
\tag{2.1}
$$

where $\kappa(\boldsymbol{x}_i, \boldsymbol{x}_j)$ is the attention score from node $i$ to node $j$ and $\boldsymbol{o}_i$ is the attention output of vertex $i$. $\sigma$ is an activation function. We denote $\boldsymbol{q}(\boldsymbol{x}) := \boldsymbol{W}_q \boldsymbol{x}, \boldsymbol{k}(\boldsymbol{x}) := \boldsymbol{W}_k \boldsymbol{x}$, and $\boldsymbol{v}(\boldsymbol{x}) := \boldsymbol{W}_v \boldsymbol{x}$ for queries, keys, and values, respectively, and $\boldsymbol{W}_q, \boldsymbol{W}_k, \boldsymbol{W}_v \in \mathbb{R}^{m \times d}$ are learnable weights. The Transformer block $\mathcal{T}$ (Vaswani et al., 2017) is defined by $\mathcal{T}(\boldsymbol{X}) := \mathrm{FFN}\left(\boldsymbol{X} + \mathrm{ATTN}(\boldsymbol{X})\right)$ where $\boldsymbol{X}$ and FFN are token embeddings and a feed-forward layer, respectively.

It is worth noting that the attention score is computed for every pair of nodes, leading to memory and computational complexity of $\mathcal{O}(N^2)$, which becomes prohibitively expensive for large-scale graphs. Many computationally efficient attention mechanisms are proposed to tackle this issue (Zaheer et al., 2020; Choromanski et al., 2021; Zhuang et al., 2023; Shirzad et al., 2023). Exphormer (Shirzad et al., 2023), a sparse GT, is specifically designed for graphs, which facilitates information exchange across real and expander edges. However, Exphormer loses its efficiency when dealing with denser graphs, as its computational complexity increases to $\mathcal{O}(N^2)$ again with the growth in graph density, significantly limiting its scalability.

### 2.2. Primal-dual relationships in kernel machines

Such quadratic complexity also exists in kernel machines, where Mercer's kernels preserve pair-wise similarities in the dual space (Mercer, 1909). For large-scale problems, it is more practical to contemplate feature representation in the primal space to circumvent quadratic complexity (Fan et al., 2008). One can refer to the representer theorem (Kimeldorf & Wahba, 1971), which delineates the optimal solution between the primal and dual spaces,

$$
\begin{aligned}
g(\boldsymbol{x}_i) &= \sum_j \alpha_j \kappa(\boldsymbol{x}_i, \boldsymbol{x}_j) = \sum_j \alpha_j \langle \boldsymbol{\phi}(\boldsymbol{x}_i), \boldsymbol{\phi}(\boldsymbol{x}_j) \rangle \\
&= \left\langle \sum_j \alpha_j \boldsymbol{\phi}(\boldsymbol{x}_j), \boldsymbol{\phi}(\boldsymbol{x}_i) \right\rangle := \langle \boldsymbol{w}, \boldsymbol{\phi}(\boldsymbol{x}_i) \rangle,
\end{aligned}
\tag{2.2}
$$

where $\alpha_j \in \mathbb{R}$ and $\boldsymbol{w} \in \mathbb{R}^p$ are variables in the dual and primal spaces. $\boldsymbol{\phi}(\cdot) : \mathbb{R}^d \to \mathbb{R}^p$ is the associated feature mapping of the kernel $\kappa$. For vector dual variables $\boldsymbol{\alpha}_j$, we can apply (2.2) to each dimension of $\boldsymbol{\alpha}_j \in \mathbb{R}^s$. Mathemat-

ically,

$$\tilde{g}(\boldsymbol{x}_i) = \sum_j \boldsymbol{\alpha}_j \kappa(\boldsymbol{x}_i, \boldsymbol{x}_j) = \sum_j \mathrm{vec}\left(\boldsymbol{\alpha}_j \boldsymbol{\phi}(\boldsymbol{x}_i)^\top \boldsymbol{\phi}(\boldsymbol{x}_j)\right)$$

$$\stackrel{(a)}{=} \left\langle \sum_j \boldsymbol{\phi}(\boldsymbol{x}_j) \otimes \boldsymbol{\alpha}_j^\top, \boldsymbol{\phi}(\boldsymbol{x}_i) \right\rangle := \langle \boldsymbol{W}, \boldsymbol{\phi}(\boldsymbol{x}_i) \rangle,$$

(2.3)

where $(a)$ comes from the vectorization property of the Kronecker product (Graham, 2018) and $\boldsymbol{W} \in \mathbb{R}^{p \times s}$. The output $\tilde{g}$ in the dual space and the attention output share a similar formulation, indicating that the attention mechanism could potentially be represented in the primal space.

## 2.3. Primphormer

However, a unique characteristic of the attention score is their asymmetry, denoted as $\kappa(\boldsymbol{x}, \boldsymbol{y}) \neq \kappa(\boldsymbol{y}, \boldsymbol{x})$, which violates the Mercer condition. Several works studied this issue and provided a mathematical foundation for allowing asymmetry as follows,

**Definition 2.1** (Asymmetric kernel trick, (Wright & Gonzalez, 2021; Lin et al., 2022; He et al., 2023a; Chen et al., 2023)). *An asymmetric kernel trick from reproducing kernel Banach spaces (RKBS) with the associated kernel function $\kappa(\cdot, \cdot) : \mathcal{X} \times \mathcal{Z} \to \mathbb{R}$ can be defined by the inner product of two real measurable feature maps from a pair of Banach spaces $\mathcal{B}_\mathcal{X}, \mathcal{B}_\mathcal{Z}$ on $\mathcal{X}, \mathcal{Z}$:*

$$\kappa(\boldsymbol{x}, \boldsymbol{z}) = \langle \boldsymbol{\phi}_q(\boldsymbol{x}), \boldsymbol{\phi}_k(\boldsymbol{z}) \rangle, \tag{2.4}$$

*where $\boldsymbol{x} \in \mathcal{X}, \boldsymbol{\phi}_q \in \mathcal{B}_\mathcal{X}, \boldsymbol{z} \in \mathcal{Z}, \boldsymbol{\phi}_k \in \mathcal{B}_\mathcal{Z}$.*

Based on (2.3) and Definition 2.1, an intuitive idea is to represent the attention output in the primal space. Chen et al. (2023) introduced a primal representation of the attention output specifically for sequence data. It collected sequence information by uniformly sampling tokens and formed a data-adaptive weight in the dual space. However, this approach has two main weaknesses. First, unlike sequences, nodes in a graph are unordered, meaning the sampling operation may break permutation equivariance, i.e., any permutation of the nodes could result in a different output. Second, the data-adaptive weight may not be sufficiently flexible, potentially limiting the model's flexibility.

To address this, we collect graph information by introducing a virtual node (Cai et al., 2023) that aggregates global information. We then formulate a new optimization problem, which forms a data-adaptive basis in the dual space:

$$\min_{\boldsymbol{\Theta}} J = \frac{1}{2} \sum_{i=1}^N \boldsymbol{e}_i^\top \boldsymbol{\Lambda} \boldsymbol{e}_i + \frac{1}{2} \sum_{j=1}^N \boldsymbol{r}_j^\top \boldsymbol{\Lambda} \boldsymbol{r}_j - \mathrm{Tr}(\boldsymbol{W}_e^\top \boldsymbol{W}_r)$$

$$\text{s.t. } \boldsymbol{e}_i = f_X \boldsymbol{W}_e \boldsymbol{\phi}_q(\boldsymbol{x}_i), i \in [N],$$
$$\boldsymbol{r}_j = f_X \boldsymbol{W}_r \boldsymbol{\phi}_k(\boldsymbol{x}_j), j \in [N],$$

(2.5)

where $\boldsymbol{\Theta} := \{\boldsymbol{W}_e, \boldsymbol{W}_r, \boldsymbol{e}_i, \boldsymbol{r}_j\}$ is the parameter set. $\boldsymbol{W}_e, \boldsymbol{W}_r \in \mathbb{R}^{N_s \times p}$, $\boldsymbol{e}_i, \boldsymbol{r}_j \in \mathbb{R}^s$, $N_s \ll N$ is a small number, and $\boldsymbol{\Lambda} \in \mathbb{R}_+^{s \times s}$ represents a diagonal regularization coefficient matrix. $\boldsymbol{\phi}_q(\cdot), \boldsymbol{\phi}_k(\cdot) : \mathbb{R}^d \to \mathbb{R}^p$ correspond to the feature maps of queries and keys, respectively. $f_X \in \mathbb{R}^{s \times N_s}$ is a data-dependent projection defined as $f_X := \boldsymbol{F} + \boldsymbol{B}\boldsymbol{X}\boldsymbol{1}_N \boldsymbol{1}_{N_s}^\top$. It serves as a virtual node that aggregates information from all nodes in the graph. Here, $\boldsymbol{F} \in \mathbb{R}^{s \times N_s}$ and $\boldsymbol{B} \in \mathbb{R}^{s \times d}$ are learnable weights.

The objective function $J$ introduces the variational principle, as discussed by Suykens (2016), which reproduces asymmetric kernels in the dual space. We propose new primal representations for graph data, i.e., $\boldsymbol{e}_i, \boldsymbol{r}_j$ in the constraints. The global aggregation $f_X$ preserves permutation equivariance. Then, we incorporate the global information into the projection weights $\boldsymbol{W}_e$ and $\boldsymbol{W}_r$, rather than into the feature mappings $\boldsymbol{\phi}_q$ and $\boldsymbol{\phi}_k$, as done by Chen et al. (2023), forming a data-adaptive basis in the dual space. The duality of the optimization problem (2.5) is given as follows,

**Theorem 2.2** (Duality). *The dual problem of the optimization (2.5) under the Karush-Kuhn-Tucker (KKT) conditions is the following linear system,*

$$\boldsymbol{K}\boldsymbol{H}_r \boldsymbol{F}_X = \boldsymbol{H}_e \boldsymbol{\Sigma},$$
$$\boldsymbol{K}^\top \boldsymbol{H}_e \boldsymbol{F}_X = \boldsymbol{H}_r \boldsymbol{\Sigma},$$

(2.6)

*which collects the solutions corresponding to the non-zero entries in $\boldsymbol{\Lambda}$ such that $\boldsymbol{\Sigma} := \boldsymbol{\Lambda}^{-1}$. $\boldsymbol{H}_e := [\boldsymbol{h}_{e_1}, \ldots, \boldsymbol{h}_{e_N}]^\top \in \mathbb{R}^{N \times s}$, and $\boldsymbol{H}_r := [\boldsymbol{h}_{r_1}, \ldots, \boldsymbol{h}_{r_N}]^\top \in \mathbb{R}^{N \times s}$ are dual variables. $\boldsymbol{K}$ corresponds to the attention score, induced by $\boldsymbol{K}_{ij} := \langle \boldsymbol{\phi}_q(\boldsymbol{x}_i), \boldsymbol{\phi}_k(\boldsymbol{x}_j) \rangle$. The proofs, Lagrangian, and KKT are provided in Appendix C.1.*

**Primal-dual relationship.** The KKT conditions (C2) yields a fact that the optimized projections $\boldsymbol{W}_r$ and $\boldsymbol{W}_e$ in the primal space are composed of all the tokens,

$$\begin{cases} \boldsymbol{W}_e = \sum_{j=1}^N f_X^\top \boldsymbol{h}_{r_j} \boldsymbol{\phi}_k(\boldsymbol{x}_j)^\top, \\ \boldsymbol{W}_r = \sum_{i=1}^N f_X^\top \boldsymbol{h}_{e_i} \boldsymbol{\phi}_q(\boldsymbol{x}_i)^\top. \end{cases} \tag{2.7}$$

By applying (2.7) to the projection scores $\boldsymbol{e}, \boldsymbol{r}$, we can formulate them in the following two ways: (a) the primal representation, and (b) the dual representation as the standard self-attention,

$$\text{Primal} : \begin{cases} \boldsymbol{e}(\boldsymbol{x}) = f_X \boldsymbol{W}_e \boldsymbol{\phi}_q(\boldsymbol{x}), \\ \boldsymbol{r}(\boldsymbol{x}) = f_X \boldsymbol{W}_r \boldsymbol{\phi}_k(\boldsymbol{x}), \end{cases}$$

$$\text{Dual} : \begin{cases} \boldsymbol{e}(\boldsymbol{x}) = \sum_{j=1}^N \tilde{\boldsymbol{h}}_{r_j} \kappa(\boldsymbol{x}, \boldsymbol{x}_j), \\ \boldsymbol{r}(\boldsymbol{x}) = \sum_{i=1}^N \tilde{\boldsymbol{h}}_{e_i} \kappa(\boldsymbol{x}_i, \boldsymbol{x}), \end{cases}$$

(2.8)

where $\boldsymbol{F}_X := f_X f_X^\top$ contains the global information, and $\tilde{\boldsymbol{h}}_{r_j} := \boldsymbol{F}_X \boldsymbol{h}_{r_j}, \tilde{\boldsymbol{h}}_{e_i} := \boldsymbol{F}_X \boldsymbol{h}_{e_i}$ are the so-called data-

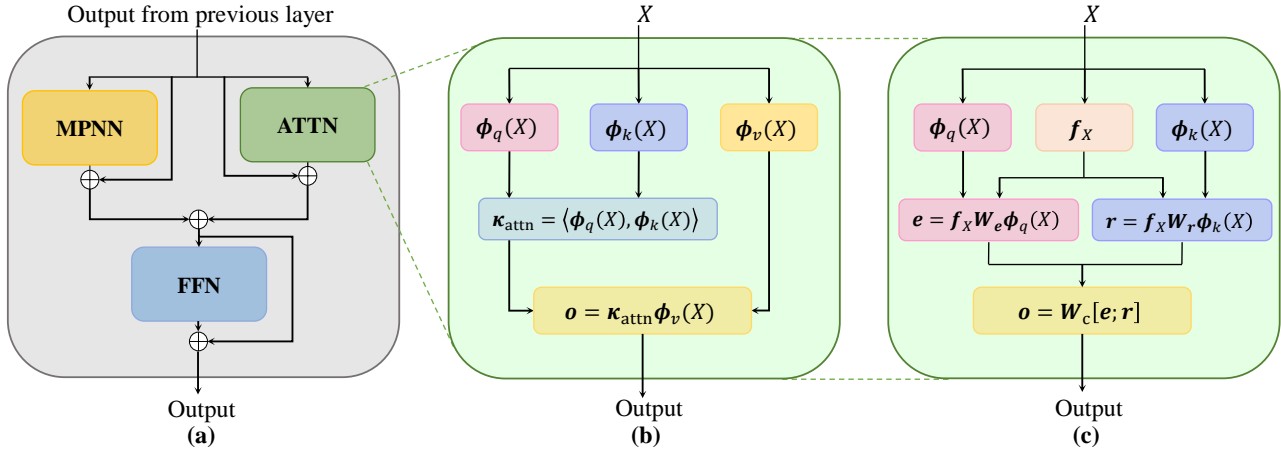

Figure 1 Illustrations of the architectures in one layer. **(a)** The GPS architecture. **(b)** The standard self-attention architecture. The attention score $\kappa_{\text{attn}}$ involves pair-wise computations. **(c)** Primphormer eliminates the need for pair-wise computations by introducing the primal representation, resulting in a new computationally efficient GT.

adaptive basis. In the primal space, we integrate token information into the projection weights $\boldsymbol{W}_r$ and $\boldsymbol{W}_e$ (2.7), representing the self-attention without pair-wise computations. The global aggregation $f_X$ inside serves as a virtual node, intended to introduce graph information to each node. Correspondingly, the attention score is computed using an asymmetric kernel trick, denoted as $\kappa(\boldsymbol{x}_i, \boldsymbol{x}_j) := \langle \boldsymbol{\phi}_q(\boldsymbol{x}_i), \boldsymbol{\phi}_k(\boldsymbol{x}_j) \rangle$, and values are the data-adaptive basis $\tilde{\boldsymbol{h}}_{r_j}, \tilde{\boldsymbol{h}}_{e_i}$, forming the self-attention in the dual space.

In contrast, Chen et al. (2023) sampled sequence information $g_X = \boldsymbol{C}\boldsymbol{X}_{\text{sub}}$ where $\boldsymbol{C} \in \mathbb{R}^{p \times d}$ and $\boldsymbol{X}_{\text{sub}}$ is uniformly sampled from $\boldsymbol{X}$, which is integrated into feature mappings, forming a data-adaptive weight $\tilde{\kappa}(\boldsymbol{x}_i, \boldsymbol{x}_j) := \langle g_X^\top \boldsymbol{\phi}_q(\boldsymbol{x}_i), g_X^\top \boldsymbol{\phi}_k(\boldsymbol{x}_j) \rangle$ and a self-attention output $\tilde{\boldsymbol{o}}(\boldsymbol{x}) = \sum_j \boldsymbol{h}_j \tilde{\kappa}(\boldsymbol{x}, \boldsymbol{x}_j)$. It is easy to check that $\tilde{\boldsymbol{o}}$ is incapable of adjusting the space that $\{\boldsymbol{h}_j\}$ spans, limiting its flexibility. Our data-adaptive basis directly adjusts the output basis thus potentially enhancing the model's flexibility.

**Model architecture.** We replace the self-attention module in the Transformer block with our primal representation (2.8) and name the resulting method Primphormer, defined as $\mathcal{T}_{\text{Pri}} := \text{FFN}\left(\boldsymbol{X} + \text{Prim}(\boldsymbol{X})\right)$. For a fair comparison, we integrate Primphormer into GPS, a powerful GT architecture that combines MPNN and Transformer blocks (Rampasek et al., 2022). The architectures are illustrated in Fig. 1, with algorithms provided in Appendix D.

**Complexity analysis.** The primal representation is a more user-friendly approach in terms of both time and memory costs. The dual representation requires $\mathcal{O}(N^2 s)$ time complexity and $\mathcal{O}(N^2 + Ns)$ memory complexity. In contrast, the primal representation only requires $\mathcal{O}(Nps)$ time complexity and $\mathcal{O}(2N_s s + 2Np)$ memory complexity with $N_s \ll N$ making an efficient self-attention mechanism feasible.

sible. The final output is obtained by concatenating two projection scores $\boldsymbol{o}(\boldsymbol{x}) := [\boldsymbol{e}(\boldsymbol{x}); \boldsymbol{r}(\boldsymbol{x})]$. To align with the user-dependent dimension $d_{\text{o}}$, a compatibility matrix $\boldsymbol{W}_c \in \mathbb{R}^{d_{\text{o}} \times 2s}$ can be further applied to the output score.

In the implementation of Primphormer, our goal is to reach the KKT points. Theorem 2.2 establishes that when the KKT conditions are met, the dual representation of Primphormer aligns with the standard self-attention formulation. However, solving the linear system (2.6) in the dual space introduces a cubic complexity. To efficiently approach the KKT points, we introduce the following lemma,

**Lemma 2.3** (Zero-valued objective with stationary solutions). *The solutions of $\boldsymbol{H}_e, \boldsymbol{H}_r, \boldsymbol{\Sigma}$ in the dual space (2.6) lead to a zero-valued objective $J$ in the primal space (2.5).*

**Implementation.** The essence of Lemma 2.3 lies in the necessity for the primal objective value to be zero under the KKT conditions, suggesting an alternative optimization approach instead of solving the dual problem. Therefore, we implement Primphormer by jointly minimizing an additional loss towards zero as follows,

$$\mathcal{L} = \mathcal{L}_{\text{task}} + \eta \sum_l J_l^2, \tag{2.9}$$

where $\eta \in \mathbb{R}_+$ is a regularization coefficient, $\mathcal{L}_{\text{task}}$ is the task-oriented loss and the final term sums up the objective loss (2.5) across layer $l$. Through regularization of this additional loss, the self-attention mechanism can be effectively represented in the primal space upon achieving a zero-valued objective.

## 3. Theoretical Results

In this section, we provide the main theorems of Primphormer. The proof details can be found in Appendix C.

## 3.1. Universal approximation

By substituting the self-attention layer with our primal representation, we obtain a new network architecture. Subsequently, the first question that intrigues us concerns the universal approximation property, delving into which functions can be uniformly approximated utilizing our network.

Here, we demonstrate that Primphormer allows universal approximation for continuous functions on both sequences and graphs. The proofs rely on a mild assumption: let feature spaces be $\mathcal{X}, \mathcal{Y} \subseteq \mathbb{R}^d$ and let $\mathcal{X}$ be a compact set. We first introduce the concept of permutation equivariance and then show that Primphormer is a universal approximator.

**Definition 3.1** (Permutation equivariance, (Hutter, 2020; Alberti et al., 2023)). *A continuous sequence-to-sequence function $f : \mathcal{X}^N \to \mathcal{Y}^N$ is equivariant to the order of elements in a sequence if for each permutation $\pi : [N] \to [N]$,*

$$f\left([\boldsymbol{x}_{\pi(1)}, \cdots, \boldsymbol{x}_{\pi(N)}]\right) = [f_{\pi(1)}(\boldsymbol{X}), \cdots, f_{\pi(N)}(\boldsymbol{X})],$$

*where $\boldsymbol{X} = [\boldsymbol{x}_1, \cdots, \boldsymbol{x}_N]$ is a sequence of $N$ tokens. Denote $f \in \mathcal{F}_{\mathrm{eq}}^N(\mathcal{X}, \mathcal{Y})$ if $f$ conforms to this definition.*

We are now ready to state the universal approximation property of Primphormer on permutation equivariant sequence-to-sequence functions.

**Theorem 3.2.** *For any function $f \in \mathcal{F}_{\mathrm{eq}}^N(\mathcal{X}, \mathcal{Y})$ and for each $\epsilon > 0$ there exists a Primphormer $\mathcal{T}_{\mathrm{Pri}}$ such that*

$$\sup_{\boldsymbol{X} \in \mathcal{X}^N} \|f(\boldsymbol{X}) - \mathcal{T}_{\mathrm{Pri}}(\boldsymbol{X})\|_\infty < \epsilon. \qquad (3.1)$$

Next, we develop the theorem for any continuous sequence-to-sequence function, stating that with a positional encoding $\boldsymbol{E} \in \mathbb{R}^{d \times N}$, a Primphormer $\mathcal{T}_{\mathrm{PE}}(\boldsymbol{X}) = \mathcal{T}_{\mathrm{Pri}}(\boldsymbol{X} + \boldsymbol{E})$ can approximate any continuous sequence-to-sequence functions on the compact domain.

**Theorem 3.3.** *For any continuous function $f : [0,1]^{d \times N} \to \mathbb{R}^{d \times N}$ and for each $\epsilon > 0$, there exists a Primphormer $\mathcal{T}_{\mathrm{PE}}$ with the positional encoding $\boldsymbol{E}$ such that*

$$\sup_{\boldsymbol{X} \in \mathcal{X}^N} \|f(\boldsymbol{X}) - \mathcal{T}_{\mathrm{PE}}(\boldsymbol{X})\|_\infty < \epsilon. \qquad (3.2)$$

Theorems 3.2, 3.3 provide universal approximation properties for functions on *sequences*. In the realm of graph-based learning, an interesting question arises: does the universality extend to functions on *graphs*?

**Universal approximator for functions on graphs**. To answer the question, we construct node and edge Primphormers on graphs. For the edge Primphormer, we consider the line graph (Cai et al., 2021) whose nodes are edges in the original graph. The edge Primphormer processes input

as a sequence of ordered pairs $((i, j), \sigma_{ij})$ where $i \le j$, $i, j \in [N]$ and an edge indicator $\sigma_{ij}$. It is evident that any permutation on these pairs describes the same graph. Considering the set of functions $f : \mathbb{R}^{N \times (N-1)} \to \mathbb{R}^{N \times (N-1)}$ with permutation equivariance, Theorem 3.2 asserts that the function $f$ can be approximated with arbitrary accuracy by Primphormer on edge input. Similarly, the node Primphormer takes an identity matrix as input and the padded adjacency matrix as a positional encoding which can be interpreted as a one-hot encoding of each node's neighbors. Considering the set of continuous functions $f : [0,1]^{N \times N} \to \mathbb{R}^{N \times N}$, Theorem 3.3 states that $f$ can be approximated as closely as desired by an appropriate Primphormer on node inputs. These ensure that Primphormer is a universal approximator for functions on graphs.

## 3.2. Expressivity

Beyond approximation theory, the second question pertains to expressivity. We demonstrate that Primphormer is as powerful as the 1-dimensional Weisfeiler-Lehman algorithm (1-WL) (Weisfeiler & Leman, 1968; Xu et al., 2019; Morris et al., 2019) in terms of distinguishing non-isomorphic graphs as follows,

**Theorem 3.4.** *Let $G = (V, E, \ell)$ be a labeled graph with $N$ nodes, and node feature matrix $\boldsymbol{X}^{(0)} := \boldsymbol{H} \in \mathbb{R}^{d \times N}$ consistent with the label $\ell$. Then, for all iterations $t \ge 0$, there exists a parameterization of Primphormer such that*

$$C_t^1(v) = C_t^1(w) \iff \boldsymbol{X}^{(t)}(v) = \boldsymbol{X}^{(t)}(w), \qquad (3.3)$$

*for all nodes $v, w \in V$, where $C_t^1 : V \to \mathbb{N}$ is the coloring function of the 1-WL test at $t$-th iteration.*

**Corollary 3.5.** *Let $G = (V, E, \ell)$ be a labeled graph with $N$ nodes, and node feature matrix $\boldsymbol{X}^{(0)} := \boldsymbol{H} \in \mathbb{R}^{d \times N}$ consistent with the label $\ell$. Then, for all iterations $t \ge 0$, there exist parameterizations of Transformer and Primphormer and a positional encoding such that*

$$\boldsymbol{X}_{\mathcal{T}}^{(t)}(v) = \boldsymbol{X}_{\mathcal{T}}^{(t)}(w) \iff \boldsymbol{X}_{\mathrm{Pri}}^{(t)}(v) = \boldsymbol{X}_{\mathrm{Pri}}^{(t)}(w), \quad (3.4)$$

*for all nodes $v, w \in V$, where $\boldsymbol{X}_{\mathcal{T}}^{(t)}$ and $\boldsymbol{X}_{\mathrm{Pri}}^{(t)}$ are node features of the output of Transformer and Primphormer models, respectively.*

These results indicate that the primal representation preserves expressivity, ensuring that Primphormer remains a powerful graph learning model.

# 4. Experimental Results

In this section, we evaluate the empirical performance of Primphormer on various graph benchmarks. To ensure diversity, datasets are collected from different sources, a detailed description of which can be found in Appendix A.

Table 1 Comparison of Primphormer with baselines on LRGB. Best results are colored in first, second, third.

| **Model** | **PascalVOC-SP** F1↑ | **COCO-SP** F1↑ | **Peptides-Func** AP↑ | **Peptides-Struct** MAE↓ | **PCQM-Contact** MRR↑ |
|---|---|---|---|---|---|
| GCN | $0.1268 \pm 0.0060$ | $0.0841 \pm 0.0010$ | $0.5930 \pm 0.0023$ | $0.3496 \pm 0.0013$ | $0.3234 \pm 0.0006$ |
| GINE | $0.1265 \pm 0.0076$ | $0.1339 \pm 0.0044$ | $0.5498 \pm 0.0079$ | $0.3547 \pm 0.0045$ | $0.3180 \pm 0.0027$ |
| GatedGCN | $0.2873 \pm 0.0219$ | $0.2641 \pm 0.0045$ | $0.5864 \pm 0.0077$ | $0.3420 \pm 0.0013$ | $0.3218 \pm 0.0011$ |
| GatedGCN+RWSE | $0.2860 \pm 0.0085$ | $0.2574 \pm 0.0034$ | $0.6069 \pm 0.0035$ | $0.3357 \pm 0.0006$ | $0.3242 \pm 0.0008$ |
| Trans.+LapPE | $0.2694 \pm 0.0098$ | $0.2618 \pm 0.0031$ | $0.6326 \pm 0.0126$ | $0.2529 \pm 0.0016$ | $0.3174 \pm 0.0020$ |
| SAN+LapPE | $0.3230 \pm 0.0039$ | $0.2592 \pm 0.0158$ | $0.6384 \pm 0.0121$ | $0.2683 \pm 0.0043$ | $0.3350 \pm 0.0003$ |
| SAN+RWSE | $0.3216 \pm 0.0027$ | $0.2434 \pm 0.0156$ | $0.6439 \pm 0.0075$ | $0.2545 \pm 0.0012$ | $0.3341 \pm 0.0006$ |
| GraphGPS | $0.3748 \pm 0.0109$ | $0.3412 \pm 0.0044$ | $0.6535 \pm 0.0041$ | $0.2500 \pm 0.0005$ | $0.3337 \pm 0.0006$ |
| Exphormer | $0.3975 \pm 0.0037$ | $0.3455 \pm 0.0009$ | $0.6527 \pm 0.0043$ | $0.2481 \pm 0.0007$ | $0.3637 \pm 0.0020$ |
| Primphormer | $0.4602 \pm 0.0077$ | $0.3903 \pm 0.0061$ | $0.6612 \pm 0.0065$ | $0.2495 \pm 0.0008$ | $0.3757 \pm 0.0079$ |

Table 2 Comparison of Primphormer with baselines on GNN benchmarks. Best results are colored in first, second, third.

| **Model** | **CIFAR10** Accuracy↑ | **MalNet-Tiny** Accuracy↑ | **MNIST** Accuracy↑ | **CLUSTER** Accuracy↑ | **PATTERN** Accuracy↑ |
|---|---|---|---|---|---|
| GCN | $55.71 \pm 0.381$ | $81.0$ | $90.71 \pm 0.218$ | $68.50 \pm 0.976$ | $71.89 \pm 0.334$ |
| GIN | $55.26 \pm 1.527$ | $88.98 \pm 0.557$ | $96.49 \pm 0.252$ | $64.72 \pm 1.553$ | $85.39 \pm 0.136$ |
| GAT | $64.22 \pm 0.455$ | $92.10 \pm 0.242$ | $95.54 \pm 0.205$ | $70.59 \pm 0.447$ | $78.27 \pm 0.186$ |
| GatedGCN | $67.31 \pm 0.311$ | $92.23 \pm 0.650$ | $97.34 \pm 0.143$ | $73.84 \pm 0.326$ | $85.57 \pm 0.088$ |
| PNA | $70.35 \pm 0.630$ | - | $97.94 \pm 0.120$ | - | - |
| DGN | $72.84 \pm 0.417$ | - | - | - | $86.68 \pm 0.034$ |
| CRaWL | $69.01 \pm 0.259$ | - | $97.94 \pm 0.050$ | - | - |
| GIN-AK+ | $72.19 \pm 0.130$ | - | - | - | $86.85 \pm 0.057$ |
| SAN | - | - | - | $76.69 \pm 0.650$ | $86.58 \pm 0.037$ |
| K-Subgraph SAT | - | - | - | $77.86 \pm 0.104$ | $86.85 \pm 0.037$ |
| EGT | $68.70 \pm 0.409$ | - | $98.17 \pm 0.087$ | $79.23 \pm 0.348$ | $86.82 \pm 0.020$ |
| GraphGPS | $72.30 \pm 0.356$ | $93.50 \pm 0.410$ | $98.05 \pm 0.126$ | $78.02 \pm 0.180$ | $86.69 \pm 0.059$ |
| Exphormer | $74.69 \pm 0.125$ | $94.02 \pm 0.209$ | $98.55 \pm 0.039$ | $78.07 \pm 0.037$ | $86.74 \pm 0.015$ |
| Primphormer | $74.13 \pm 0.241$ | $93.62 \pm 0.242$ | $98.56 \pm 0.042$ | $78.01 \pm 0.162$ | $86.68 \pm 0.056$ |

In particular, we conducted experiments on the benchmark datasets including the image-based graph datasets CIFAR10, MNIST, COCO-SP, and PascalVOC-SP; the synthetic SBM datasets PATTERN and CLUSTER; the code graph dataset MalNet-Tiny; the molecular datasets including Peptides-Func, Peptides-Struct, and PCQM-Contact (Dwivedi et al., 2022a; Freitas et al., 2021; Dwivedi et al., 2022b; 2023); the large-scale ogbn-products dataset (Hu et al., 2020), and the graph isomorphism benchmark BREC (Wang & Zhang, 2024). In our experiments, we use feature maps defined as $\phi_q(\boldsymbol{x}) := \boldsymbol{q}(\boldsymbol{x})/\|\boldsymbol{q}(\boldsymbol{x})\|_2$ and $\phi_k(\boldsymbol{x}) := \boldsymbol{k}(\boldsymbol{x})/\|\boldsymbol{k}(\boldsymbol{x})\|_2$ as used by Chen et al. (2023).

**Long-range graph benchmark (LRGB).** We conducted experiments on LRGB (Dwivedi et al., 2022b) to evaluate the models' capabilities in learning long-range dependencies within input graphs. Table 1 presents the results of Primphormer with several baselines. Our approach outperforms the baselines on four of the five datasets while showing competitive performance on the rest of the datasets.

**GNN benchmark datasets.** We also evaluate our method with broader baselines on graph benchmark datasets, namely CIFAR10, MNIST, CLUSTER, PATTERN, and the code graph dataset MalNet-Tiny (Dwivedi et al., 2023; Freitas et al., 2021), as reported in Table 2. It is observed that Primphormer outperforms MNIST and ranks as the second-best approach on two additional datasets, showcasing its strong performance across various dataset types.

**Efficiency validation.** Primphormer leverages the primal representation for GTs to reduce the computational burden. As the aforementioned results demonstrate the promising performance of Primphormer, we further validate its efficiency by comparing it to other computationally efficient attention mechanisms within the GPS architecture (Rampasek et al., 2022). The selected mechanisms include linear attention models BigBird (Zaheer et al., 2020) and Performer (Choromanski et al., 2021), a sparse attention mechanism, Exphormer (Shirzad et al., 2023), the sequence-specific Primal-Atten (Chen et al., 2023), and the full at-

Table 3 Comparison of attentions in GPS. Best results are colored in **first**, **second**, **third**. OOM means out of memory.

| MODEL
GPS | CIFAR10
ACCURACY↑ | MALNET-TINY
ACCURACY↑ | PASCALVOC-SP
F1↑ | PEPTIDES-FUNC
AP↑ | OGBN-PRODUCTS
ACCURACY↑ |
|---|---|---|---|---|---|
| MPNN-ONLY | 69.95 ± 0.499 | 92.23 ± 0.650 | 0.3016 ± 0.0031 | 0.6159 ± 0.0048 | 74.25 ± 0.214s |
| +TRANSFORMER | 72.31 ± 0.344 | 93.50 ± 0.410 | 0.3748 ± 0.0109 | 0.6535 ± 0.0041 | OOM |
| +BIGBIRD | 70.48 ± 0.106 | 92.34 ± 0.340 | 0.2762 ± 0.0069 | 0.5854 ± 0.0079 | 73.82 ± 0.412 |
| +PERFORMER | 70.67 ± 0.338 | 92.64 ± 0.780 | 0.3724 ± 0.0131 | 0.6475 ± 0.0056 | 74.30 ± 0.211 |
| +PRIM-ATTEN | 71.57 ± 0.256 | 92.97 ± 0.228 | 0.3173 ± 0.0055 | 0.6447 ± 0.0046 | 74.47 ± 0.134 |
| +EXPHORMER | 74.69 ± 0.125 | 94.02 ± 0.209 | 0.3975 ± 0.0037 | 0.6527 ± 0.0043 | 74.67 ± 0.179 |
| +PRIMPHORMER | 74.13 ± 0.241 | 93.62 ± 0.242 | 0.4602 ± 0.0077 | 0.6612 ± 0.0065 | 74.89 ± 0.281 |

Table 4 Efficiency comparisons on running time and peak memory consumption.

| MODEL
GPS | TIME (S/EPOCH) | | | | | PEAK MEMORY USAGE (GB) | | | | |
|---|---|---|---|---|---|---|---|---|---|---|
| | CIFAR. | MALNET. | PASCAL. | FUNC. | PROD. | CIFAR. | MALNET. | PASCAL. | FUNC. | PROD. |
| MPNN-ONLY | 20.3 | 24.5 | 15.7 | 4.8 | 21.1 | 2.31 | 1.92 | 4.18 | 2.45 | 11.97 |
| +TRANSFORMER | 28.0 | 232.4 | 35.6 | 12.8 | - | 3.81 | 35.32 | 7.82 | 8.46 | OOM |
| +BIGBIRD | 55.2 | 325.6 | 52.3 | 51.9 | 93.9 | 2.81 | 2.71 | 4.99 | 4.99 | 17.29 |
| +PERFORMER | 50.8 | 73.5 | 49.7 | 21.7 | 22.7 | 10.5 | 11.59 | 6.14 | 7.71 | 16.14 |
| +PRIM-ATTEN | 32.1 | 62.5 | 25.7 | 7.9 | 22.6 | 2.74 | 2.58 | 4.74 | 3.38 | 13.63 |
| +EXPHORMER | 44.5 | 62.1 | 35.2 | 7.6 | 25.4 | 5.54 | 10.38 | 7.35 | 4.81 | 31.09 |
| +PRIMPHORMER | 32.6 | 61.9 | 25.3 | 7.7 | 22.1 | 2.74 | 2.86 | 4.72 | 3.41 | 13.35 |

tention mechanism. We conduct the experiments on CIFAR10, MalNet-Tiny, PascalVOC, Peptides-Func, and a large-scale graph ogbn-products. Since ogbn-products is too large to be loaded into GPU, we use the random partitioning method previously used by Wu et al. (2022; 2023a).

As shown in Table 3, Primphormer demonstrates superior performance over other attention mechanisms such as BigBird, Performer, and Prim-Atten, while also exhibiting competitive performance with Exphormer. Table 4 presents a comparison of running time and peak memory usage across different methods. Primphormer demonstrates superior performance in both running time and memory consumption compared to other approaches. For example, in the MalNet-Tiny dataset, linear attention mechanisms introduce significant computational overhead. While Prim-Atten offers good efficiency, its performance on graph tasks lags due to its sequence-specific nature. Both Primphormer and Exphormer, designed for graphs, exhibit similar running times. Nevertheless, Primphormer consumes less memory as its complexity depends solely on the number of nodes, whereas Exphormer's complexity is controlled by the number of nodes and edges. In the ogbn-products dataset, which comprises approximately 2 million nodes and 61 million edges, Primphormer showcases the most efficient results compared with other methods. In summary, our experiments demonstrate that Primphormer exhibits competitive performance while maintaining user-friendly memory and computational costs.

**Expressivity Tests.** We evaluate the expressive power of our approach on the BREC benchmark (Wang & Zhang, 2024), as shown in Tab. 5. For reference, we report the results of Graphormer (Ying et al., 2021) and APE-GT(Black et al., 2024) as graph Transformer baselines, and 3-WL (Müller & Morris, 2024), which serves as a potential expressivity upper-bound. We compare pure Primphormer to the standard Transformer (Vaswani et al., 2017) and Prim-Atten (Chen et al., 2023) using two positional encodings: LAP/LapPE (Kreuzer et al., 2021) and SPE (Huang et al., 2024), all evaluated without the MPNN layer. We find that both Transformer and Primphormer outperform Prim-Atten. And results show that Primphormer and the standard Transformer achieve similar performance, consistent with our theoretical results discussed in Sec. 3.2.

Table 5 Results on the BREC benchmark. Basic, Regular, Extension, and CFI are subsets of the BREC benchmark. Experiments are averaged over 5 runs.

| MODEL | PE | BAS.↑ | REG.↑ | EXT.↑ | CFI↑ | ALL↑ |
|---|---|---|---|---|---|---|
| GRAPHORMER | | 16 | 12 | 41 | 10 | 79 |
| APE-GT | | 50.6 | 31.3 | 62.4 | 1 | 145.3 |
| 3-WL | | 60 | 50 | 100 | 60 | 270 |
| TRANSFORMER | | 47.2 | 39 | 65.2 | 3 | 154.4 |
| PRIM-ATTEN | LAP | 12.8 | 19 | 13.6 | 0.6 | 46 |
| PRIMPHORMER | | 51.6 | 42 | 72.4 | 3 | 169 |
| TRANSFORMER | | 59.8 | 49.4 | 98.6 | 5.2 | 213 |
| PRIM-ATTEN | SPE | 46.4 | 49 | 73 | 3 | 171.4 |
| PRIMPHORMER | | 60 | 50 | 100 | 9.4 | 219.4 |

## 5. Related Work

**Graph Transformers.** Transformers have demonstrated success in natural language processing (Vaswani et al., 2017) and computer vision tasks (Liu et al., 2021). Recently, researchers have explored the application of Transformers in graph representation learning to address issues such as over-smoothing (Nguyen et al., 2023) and over-squashing (Giraldo et al., 2023) observed in MPNNs. Graph Transformers operate on a fully connected graph where nodes are pairwise connected, encoding the original graph structure into positional encodings. Spectral Attention Networks (SAN) (Kreuzer et al., 2021) introduced conditional attention for both real and virtual edges and implemented Laplacian positional encoding for nodes. Graphormer (Ying et al., 2021) and GraphiT (Mialon et al., 2021) incorporated relative positional encodings based on pairwise graph distances and diffusion kernels, respectively. GPS proposed a framework that combined MPNNs with attention mechanisms (Rampasek et al., 2022).

The quadratic complexity in traditional GTs has motivated the development of computationally efficient attention mechanisms. Linear attention mechanisms, such as NodeFormer (Wu et al., 2022) and SGFormer (Wu et al., 2023b), aim to reduce computational complexity by decomposing or approximating the kernel matrix, operating in the dual space. For example, NodeFormer uses a random feature-based approach, while SGFormer drops the softmax activation to approximate the kernel matrix. Difformer (Wu et al., 2023a) introduced a diffusion-based Transformer model with linear complexity, although their attention mechanisms are limited to nodes in randomly sampled mini-batches. Another strategy is the sparse Transformer, which enhances computational efficiency by restricting node interactions. Exphormer (Shirzad et al., 2023) limited interactions across real and expander edges, achieving linear complexity to the number of nodes and edges. However, the efficiency of Exphormer diminishes as graphs become denser. A survey on efficient Transformers is given by Fournier et al. (2023).

**Primal and dual representations.** The quadratic complexity also arises in kernel machines in duality and can be circumvented by transferring a dual problem to its primal form. Models such as the support vector machine (Cortes & Vapnik, 1995), least squares support vector machine (Suykens & Vandewalle, 1999), and kernel principal component analysis (Mika et al., 1999) exhibit this characteristic. The associated pair-wise kernels are symmetric and positive-definite, whereas attention scores are inherently asymmetric, violating the Mercer condition (Mercer, 1909). Recent research has explored a new primal-dual perspective to accommodate such asymmetry in kernel machines. To incorporate asymmetric kernel functions, Lin et al. (2022) proposed an asymmetric kernel trick from a pair of RKBSs. He et al. (2023b) converted an asymmetric kernel to a complex-valued Hermitian function by the magnetic transform. Suykens (2016) introduced a novel variational principle to dissect the primal-dual relationship concerning the singular value decomposition of an asymmetric kernel matrix, a concept further extended to classification tasks by He et al. (2023a). This variational principle was also leveraged by Chen et al. (2023) to interpret attention mechanisms in sequences. However, due to the distinctions between sequences and graphs, this model is unsuitable for graph-based learning.

## 6. Conclusion

In this paper, we propose Primphormer, a new framework for graph Transformers. Primphormer models the self-attention mechanism on graphs in the primal space, avoiding pair-wise computations, which enables an efficient variant of graph Transformers. Our primal-dual analysis shows that Primphormer can be implemented by introducing an additional primal objective loss. Due to its efficiency in both runtime and memory storage, Primphormer has the potential to support larger and deeper neural networks and enable larger batch sizes, enhancing model capacity and generalization ability. Primphormer also benefits from the universal approximation property for functions on both sequences and graphs, potentially possessing strong generalization capabilities to unseen data or tasks. We exhibit that its expressive power is as powerful as Transformers in distinguishing non-isomorphic graphs, showing that the primal representation can preserve expressivity. Experimental results on various graph benchmarks demonstrate the effectiveness and efficiency of the proposed Primphormer.

An interesting avenue for future work is exploring how edge features can be incorporated into Primphormer's structure. Edge features can be added to attention scores in an entry-wise manner as data-adaptive kernels (Liu et al., 2020). Exploring the primal representation of these kernels allows us to incorporate edge information into attention mechanisms, potentially resulting in a stronger GT. Additionally, fine-tuning schemes like LoRA (Hu et al., 2022) are promising for large models. Studying LoRA from a primal-dual perspective may lead to more efficient fine-tuning methods.

## Impact Statement

This paper presents work whose goal is to advance the field of Machine Learning. There are many potential societal consequences of our work, none of which we feel must be specifically highlighted here.

## Acknowledgments

The authors would like to thank the anonymous reviewers for their insightful comments.

The research leading to these results has received funding from the National Key Research Development Project (2023YFF1104202) and the National Natural Science Foundation of China (62376155).

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

# Appendix

## A. Data Descriptions

Here, we introduce the datasets in the experiments. A summary of the dataset statistics is shown in Tab. A1.

**CIFAR10 and MNIST.** CIFAR10 and MNIST are the graph equivalents of the image classification datasets of the same name. A graph is created by constructing the 8-nearest neighbor graph of the SLIC superpixels of the image. These are both 10-class graph classification problems (Dwivedi et al., 2023).

**PascalVOC-SP and COCO-SP.** These are similar graph versions of image datasets, but they are larger images and the task is to perform node classification, i.e., semantic segmentation of super-pixels. These graphs are larger, and the tasks are more complex than CIFAR10 and MNIST (Dwivedi et al., 2022a).

**CLUSTER and PATTERN.** PATTERN and CLUSTER are node classification problems. Both are synthetic datasets that are sampled from a Stochastic Block Model (SBM), which is a popular way to model communities. In PATTERN, the prediction task is to identify if a node belongs to one of the 100 possible predetermined sub-graph patterns. In CLUSTER, the goal is to classify nodes into six different clusters with the same distribution (Dwivedi et al., 2023).

**MalNet-Tiny.** Malnet-Tiny is a smaller dataset generated from a larger dataset for identifying malware based on function call graphs from Android APKs. The tiny dataset contains 5000 graphs, each with up to 5000 nodes. The task is to predict the graph as being benign or from one of four types of malware (Freitas et al., 2021).

**Peptides-Func, Peptides-Struct, and PCQM-Contact.** These datasets are molecular graphs introduced as a part of the Long Range Graph Benchmark (LRGB). On PCQM-Contact, the task is edge-level, and we need to rank the edges. Peptides-Func is a multi-label graph classification task with 10 labels. Peptides-Struct is a graph-level regression of 11 structural properties of the molecules (Dwivedi et al., 2022a;b).

**OGBN-products.** The ogbn-products dataset is an undirected and unweighted graph, representing an Amazon product co-purchasing network. Nodes represent products sold on Amazon, and edges between two products indicate that the products are purchased together. Specifically, node features are generated by extracting bag-of-words features from the product descriptions followed by a Principal Component Analysis to reduce the dimension to 100. The task is to predict the category of a product in a multi-class classification setup, where the 47 top-level categories are used for target labels (Hu et al., 2020). We use the random partitioning method with ten partitions as previously utilized in Wu et al. (2022; 2023a).

**BREC.** BREC is a dataset for GNN expressiveness comparison. It addresses the limitations of previous datasets, including difficulty, granularity, and scale, by incorporating 400 pairs of various graphs in four categories (Basic, Regular, Extension, and CFI). The graphs are organized pair-wise, where each pair is tested individually to return whether a GNN can distinguish them. We use the evaluation method, RPC (Reliable Paired Comparisons), with a contrastive training framework as introduced in Wang & Zhang (2024).

Table A1 Dataset statistics.

| DATASET | GRAPHS | AVG. NODES | AVG.EDGES | TASK LEVEL | CLASS | METRIC |
|---------|--------|-----------|-----------|-----------|-------|--------|
| MNIST | 70,000 | 70.6 | 564.5 | GRAPH | 10 | ACC |
| CIFAR10 | 60,000 | 117.6 | 941.1 | GRAPH | 10 | ACC |
| PATTERN | 14,000 | 118.9 | 3039.3 | INDUCTIVE NODE | 2 | ACC |
| CLUSTER | 12,000 | 117.2 | 2150.9 | INDUCTIVE NODE | 6 | ACC |
| MALNET-TINY | 5,000 | 1,410.3 | 2,859.9 | GRAPH | 5 | ACC |
| PASCALVOC-SP | 11,355 | 479.4 | 2710.5 | INDUCTIVE NODE | 21 | F1 |
| COCO-SP | 123,286 | 476.9 | 2710.5 | INDUCTIVE NODE | 81 | F1 |
| PCQM-CONTACT | 529,434 | 30.1 | 61.0 | INDUCTIVE LINK | LINK RANKING | MRR |
| PEPTIDES-FUNC | 15,535 | 150.9 | 307.3 | GRAPH | 10 | AP |
| PEPTIDES-STRUCT | 15,535 | 150.9 | 309.3 | GRAPH | 11 | MAE |
| OGBN-PRODUCTS | 1 | 2,449,029 | 61,859,140 | NODE | 47 | ACC |

# B. Hyperparameters

Our selection of hyperparameters was guided by the instructions in GPS (Rampasek et al., 2022) and Exphormer (Shirzad et al., 2023). Further details can be found in Tables. A3- A4.

In our model, we introduced additional hyperparameters, the dimensions of the data-dependent projection, denoted as $N_s$ and its low rank $s$, and the regularization coefficient $\eta$. We utilized grid search to explore these hyperparameters across $N_s, s \in \{20, 30, 40, 50, 60\}$, and $\eta \in \{0.1, 0.01\}$. For the remaining hyperparameters, we conducted a linear search for each parameter to determine the best values. Throughout all experiments, we employed CustomGatedGCN as the MPNN module alongside Primphormer except for ogbn-products dataset where we use GCN. To ensure fair comparisons, we maintained a similar parameter budget to that of GraphGPS.

Table A4 presents the hyperparameters used in our efficiency experiments. To maintain consistency in our evaluations of various attention mechanisms, we applied the same parameters for a fair comparison.

Table A2 Hyperparameters used in Primphormer for datasets: PascalVOC-SP, COCO-SP, Peptides-Func, Peptides-Struct, PCQM-Contact.

| HYPERPARMETER | PASCALVOC-SP | COCO-SP | PEPTIDES-FUNC | PEPTIDES-STRUCT | PCQM-CONTACT |
|---|---|---|---|---|---|
| #LAYERS | 6 | 7 | 4 | 4 | 7 |
| HIDDEN DIM | 80 | 56 | 96 | 96 | 64 |
| #HEADS | 1 | 2 | 4 | 4 | 4 |
| DROPOUT | 0.15 | 0.0 | 0.1 | 0.15 | 0.0 |
| ATTENTION DROPOUT | 0.5 | 0.5 | 0.1 | 0.5 | 0.56 |
| PE | LAPPE | LAPPE | RWSE | RWSE | LAPPE |
| PE DIM | 16 | 16 | 16 | 20 | 16 |
| BATCH SIZE | 200 | 150 | 200 | 200 | 128 |
| LEARNING RATE | 1E-3 | 1E-3 | 1E-3 | 1E-3 | 3E-4 |
| #EPOCHS | 300 | 300 | 250 | 250 | 250 |
| WEIGHT DECAY | 1E-5 | 1E-2 | 1E-2 | 1E-2 | 0.0 |
| $N_s$ | 30 | 20 | 30 | 40 | 30 |
| $\eta$ | 0.1 | 0.1 | 0.1 | 0.1 | 0.1 |
| $s$ | 30 | 20 | 30 | 40 | 30 |
| #PARAMETERS | 508305 | 315305 | 470693 | 468783 | 386526 |

Table A3 Hyperparameters used in Primphormer for datasets: CIFAR10, MNIST, MalNet-Tiny, PATTERN, CLUSTER, BREC.

| HYPERPARMETER | CIFAR10 | MNIST | MALNET-TINY | PATTERN | CLUSTER | BREC |
|---|---|---|---|---|---|---|
| #LAYERS | 3 | 4 | 5 | 6 | 12 | 5 |
| HIDDEN DIM | 52 | 40 | 84 | 48 | 52 | 32 |
| #HEADS | 1 | 1 | 1 | 1 | 1 | 4 |
| DROPOUT | 0.15 | 0.1 | 0.15 | 0.0 | 0.15 | 0.0 |
| ATTENTION DROPOUT | 0.5 | 0.5 | 0.5 | 0.5 | 0.5 | 0.5 |
| PE | ESLAPPE | ESLAPPE | - | ESLAPPE | ESLAPPE | LAPPE/SPE |
| PE DIM | 8 | 8 | - | 8 | 10 | 16 |
| BATCH SIZE | 200 | 200 | 64 | 128 | 48 | 16 |
| LEARNING RATE | 1E-3 | 1E-3 | 1E-3 | 1E-3 | 1E-3 | 1E-3 |
| #EPOCHS | 300 | 300 | 300 | 200 | 300 | 25 |
| WEIGHT DECAY | 1E-2 | 1E-5 | 1E-3 | 1E-5 | 1E-5 | 1E-2 |
| $N_s$ | 20 | 30 | 50 | 30 | 40 | 20 |
| $\eta$ | 0.1 | 0.1 | 0.1 | 0.1 | 0.1 | 0.01 |
| $s$ | 20 | 30 | 50 | 30 | 40 | 20 |
| #PARAMETERS | 112957 | 101714 | 519605 | 208387 | 499386 | 52238 |

Table A4 Hyperparameters used in Table. 4.

| HYPERPARMETER | CIFAR10 | MALNET-TINY | PASVALVOC-SP | PEPTIDES-FUNC | OGBN-PRODUCTS |
|---|---|---|---|---|---|
| #LAYERS | 5 | 5 | 4 | 4 | 2 |
| HIDDEN DIM | 40 | 64 | 96 | 96 | 128 |
| BATCH SIZE | 128 | 4 | 32 | 128 | - |

# C. Proofs of theoretical results

In this section, we provide the proof of theoretical results in this paper.

## C.1. Proof details of Theorem 2.2

The Lagrangian of (2.5) is defined by,

$$
\mathcal{L}(\boldsymbol{W}_e, \boldsymbol{W}_r, \boldsymbol{e}_i, \boldsymbol{r}_j, \boldsymbol{h}_{e_i}, \boldsymbol{h}_{r_j}) = \frac{1}{2} \sum_{i=1}^{N} \boldsymbol{e}_i^\top \boldsymbol{\Lambda} \boldsymbol{e}_i + \frac{1}{2} \sum_{j=1}^{N} \boldsymbol{r}_j^\top \boldsymbol{\Lambda} \boldsymbol{r}_j - \mathrm{Tr}(\boldsymbol{W}_e^\top \boldsymbol{W}_r)
$$
$$
- \sum_{i=1}^{N} \boldsymbol{h}_{e_i}^\top (\boldsymbol{e}_i - f_X \boldsymbol{W}_e \phi_q(\boldsymbol{x}_i)) - \boldsymbol{h}_{r_j}^\top (\boldsymbol{r}_j - f_X \boldsymbol{W}_r \phi_k(\boldsymbol{x}_j)),
$$

(C1)

where $\boldsymbol{h}_{e_i}, \boldsymbol{h}_{r_j} \in \mathbb{R}^s$ are dual variable vectors corresponding to the equality constraints regarding the projection scores $\boldsymbol{e}_i$ and $\boldsymbol{r}_j$.

By taking the partial derivatives to the Lagrangian (C1), the Karush-Kuhn-Tucker (KKT) conditions lead to the following equalities,

$$
\begin{cases}
\dfrac{\partial \mathcal{L}}{\partial \boldsymbol{W}_e} = 0 \Rightarrow \boldsymbol{W}_r = \sum_{i=1}^{N} f_X^\top \boldsymbol{h}_{e_i} \phi_q(\boldsymbol{x}_i)^\top \\[2mm]
\dfrac{\partial \mathcal{L}}{\partial \boldsymbol{W}_r} = 0 \Rightarrow \boldsymbol{W}_e = \sum_{j=1}^{N} f_X^\top \boldsymbol{h}_{r_j} \phi_k(\boldsymbol{x}_j)^\top \\[2mm]
\dfrac{\partial \mathcal{L}}{\partial \boldsymbol{e}_i} = 0 \Rightarrow \boldsymbol{\Lambda} \boldsymbol{e}_i = \boldsymbol{h}_{e_i}, \quad i \in [N] \\[2mm]
\dfrac{\partial \mathcal{L}}{\partial \boldsymbol{r}_j} = 0 \Rightarrow \boldsymbol{\Lambda} \boldsymbol{r}_j = \boldsymbol{h}_{r_j}, \quad j \in [N] \\[2mm]
\dfrac{\partial \mathcal{L}}{\partial \boldsymbol{h}_{e_i}} = 0 \Rightarrow \boldsymbol{e}_i = f_X \boldsymbol{W}_e \phi_q(\boldsymbol{x}_i), \quad i \in [N] \\[2mm]
\dfrac{\partial \mathcal{L}}{\partial \boldsymbol{h}_{r_j}} = 0 \Rightarrow \boldsymbol{r}_j = f_X \boldsymbol{W}_r \phi_k(\boldsymbol{x}_j), \quad j \in [N].
\end{cases}
$$

(C2)

By eliminating the primal variables $\boldsymbol{W}_e$ and $\boldsymbol{W}_r$, we have,

$$
\begin{cases}
\sum_{j=1}^{N} \boldsymbol{F}_X \boldsymbol{h}_{r_j} \phi_k(\boldsymbol{x}_j)^\top \phi_q(\boldsymbol{x}_i) = \boldsymbol{\Lambda}^{-1} \boldsymbol{h}_{e_i}, \quad i \in [N], \\[2mm]
\sum_{i=1}^{N} \boldsymbol{F}_X \boldsymbol{h}_{e_i} \phi_q(\boldsymbol{x}_i)^\top \phi_k(\boldsymbol{x}_j) = \boldsymbol{\Lambda}^{-1} \boldsymbol{h}_{r_j}, \quad j \in [N],
\end{cases}
$$

(C3)

where $\boldsymbol{F}_X := f_X f_X^\top \in \mathbb{S}_+^{s \times s}$ is the auto-correlation matrix. It can be expressed in the following matrix form,

$$
\begin{bmatrix} \boldsymbol{0}_{N \times N} & [\phi_q(\boldsymbol{x}_i)^\top \phi_k(\boldsymbol{x}_j)] \\ [\phi_k(\boldsymbol{x}_j)^\top \phi_q(\boldsymbol{x}_i)] & \boldsymbol{0}_{N \times N} \end{bmatrix} \begin{bmatrix} \boldsymbol{H}_e \\ \boldsymbol{H}_r \end{bmatrix} \boldsymbol{F}_X = \begin{bmatrix} \boldsymbol{H}_e \\ \boldsymbol{H}_r \end{bmatrix} \boldsymbol{\Lambda}^{-1},
$$

(C4)

with $\boldsymbol{H}_e := [\boldsymbol{h}_{e_1}, \dots, \boldsymbol{h}_{e_N}]^\top \in \mathbb{R}^{N \times s}$, and $\boldsymbol{H}_r := [\boldsymbol{h}_{r_1}, \dots, \boldsymbol{h}_{r_N}]^\top \in \mathbb{R}^{N \times s}$.

Then it can be noticed that the optimization problem (2.5) in the dual space yields the following generalized eigenvalue problem with an asymmetric kernel $K$,

$$KH_rF_X = H_e\Sigma,$$
$$K^\top H_eF_X = H_r\Sigma, \tag{C5}$$

which collects the solutions corresponding to the non-zero entries in $\Lambda$ such that $\Sigma := \Lambda^{-1}$. The asymmetric kernel matrix $K$, induced by $K_{ij} := \langle \phi_q(x_i), \phi_k(x_j) \rangle, \forall i, j \in [N]$, corresponds to the attention matrix.

## C.2. Derivation of scores (2.8) in the primal and dual spaces

With the derivations and KKT conditions of the primal-dual optimization above, the primal and dual representation for self-attention can be formulated as follows,

$$\text{Primal}: \begin{cases} e(x) = f_X W_e \phi_q(x), \\ r(x) = f_X W_r \phi_k(x). \end{cases} \tag{C6}$$

$$\text{Dual}: \begin{cases} e(x) = f_X W_e \phi_q(x_i) = \sum_{j=1}^{N} F_X h_{r_j} \phi_k(x_j)^\top \phi_q(x), \\ r(x) = f_X W_r \phi_k(x_i) = \sum_{i=1}^{N} F_X h_{e_i} \phi_q(x_i)^\top \phi_k(x). \end{cases} \tag{C7}$$

Then, the primal and dual representations for self-attention can be formulated as follows,

$$\text{Primal}: \begin{cases} e(x) = W_{e|X}^\top \phi_q(x), \\ r(x) = W_{r|X}^\top \phi_k(x), \end{cases} \quad \text{Dual}: \begin{cases} e(x) = \sum_{j=1}^{N} \tilde{h}_{r_j} \kappa(x, x_j), \\ r(x) = \sum_{i=1}^{N} \tilde{h}_{e_i} \kappa(x_i, x), \end{cases} \tag{C8}$$

where $W_{e|X}^\top := f_X W_e \in \mathbb{R}^{s \times p}$, $W_{r|X}^\top := f_X W_r \in \mathbb{R}^{s \times p}$ and $\tilde{h}_{r_j} := F_X h_{r_j}, \tilde{h}_{e_i} := F_X h_{e_i}$ are values for self-attention, respectively.

## C.3. Proof details of Lemma 2.3

*Proof.* Based on the KKT conditions (C2) and (2.6), the objective on stationary points is,

$$\begin{aligned} J &= \frac{1}{2} \sum_{i=1}^{N} e_i^\top \Lambda e_i + \frac{1}{2} \sum_{j=1}^{N} r_j^\top \Lambda r_j - \text{Tr}\left(W_e^\top W_r\right) \\ &= \frac{1}{2} \sum_{i=1}^{N} \left(\Lambda^{-1} h_{e_i}\right)^\top \Lambda \Lambda^{-1} h_{e_i} + \frac{1}{2} \sum_{j=1}^{N} \left(\Lambda^{-1} h_{r_j}\right)^\top \Lambda \Lambda^{-1} h_{r_j} \\ &\quad - \text{Tr}\left(\left(\sum_{j=1}^{N} \phi_k(x_j) h_{r_j}^\top f_X\right) \cdot \left(\sum_{i=1}^{N} f_X^\top h_{e_i} \phi_q(x_i)^\top\right)\right) \\ &= \frac{1}{2} \sum_{i=1}^{N} h_{e_i}^\top \Lambda^{-1} h_{e_i} + \frac{1}{2} \sum_{j=1}^{N} h_{r_j}^\top \Lambda^{-1} h_{r_j} - \text{Tr}\left(\sum_{i,j} \phi_k(x_j) h_{r_j}^\top F_X h_{e_i} \phi_q(x_i)^\top\right) \\ &= \frac{1}{2}\text{Tr}\left(H_e \Sigma H_e^\top\right) + \frac{1}{2}\text{Tr}\left(H_r \Sigma H_r^\top\right) - \text{Tr}\left(\sum_{i,j} \phi_q(x_i)^\top \phi_k(x_j) h_{r_j}^\top F_X h_{e_i}\right) \\ &= \frac{1}{2}\text{Tr}\left(H_e \Sigma H_e^\top\right) + \frac{1}{2}\text{Tr}\left(H_r \Sigma H_r^\top\right) - \text{Tr}\left(K H_r F_X H_e^\top\right) \\ &= \frac{1}{2}\text{Tr}\left(K H_r F_X H_e^\top\right) + \frac{1}{2}\text{Tr}\left(K^\top H_e F_X H_r^\top\right) - \text{Tr}\left(K H_r F_X H_e^\top\right) \\ &= \frac{1}{2}\text{Tr}\left(K^\top H_e F_X H_r^\top\right) - \frac{1}{2}\text{Tr}\left(K H_r F_X H_e^\top\right) \\ &= \frac{1}{2}\text{Tr}\left(H_e F_X H_r^\top K^\top\right) - \frac{1}{2}\text{Tr}\left(K H_r F_X H_e^\top\right) \\ &= \frac{1}{2}\text{Tr}\left(\left(H_e F_X H_r^\top K^\top\right)^\top\right) - \frac{1}{2}\text{Tr}\left(K H_r F_X H_e^\top\right) \\ &= \frac{1}{2}\text{Tr}\left(K H_r F_X H_e^\top\right) - \frac{1}{2}\text{Tr}\left(K H_r F_X H_e^\top\right) = 0. \end{aligned} \tag{C9}$$

$\square$

### C.4. Proof details of Theorem 3.2

*Proof.* The proof follows ideas in (Alberti et al., 2023). We first introduce the Sumformer $\mathcal{S}$ and we divide the approximation into two parts: 1) approximate $f$ by a $\mathcal{S}$ and 2) approximate $\mathcal{S}$ by a Primphormer $\mathcal{T}_{\mathrm{Pri}}$.

**Definition C.1** (Sumformer). Let $d' \in \mathbb{N}$ and let there be two functions $\boldsymbol{\xi} : \mathcal{X} \to \mathbb{R}^{d'}, \psi : \mathcal{X} \times \mathbb{R}^{d'} \to \mathcal{Y}$. A Sumformer is a sequence-to-sequence function $\mathcal{S} : \mathcal{X}^N \to \mathcal{Y}^N$ which is evaluated by first computing

$$\boldsymbol{\Xi} := \sum_{k=1}^{N} \boldsymbol{\xi}(\boldsymbol{x}_k), \tag{C10}$$

and then

$$\mathcal{S}\left([\boldsymbol{x}_1, \cdots, \boldsymbol{x}_N]\right) := \left[\psi(\boldsymbol{x}_1, \boldsymbol{\Xi}), \cdots, \psi(\boldsymbol{x}_N, \boldsymbol{\Xi})\right]. \tag{C11}$$

**Lemma C.2** (Alberti et al. (2023), universal approximation of Sumformer). *For each function $f \in \mathcal{F}_{\mathrm{eq}}^N(\mathcal{X}, \mathcal{Y})$ and for each $\epsilon > 0$ there exists a Sumformer $\mathcal{S}$ such that*

$$\sup_{\boldsymbol{X} \in \mathcal{X}^N} \|f(\boldsymbol{X}) - \mathcal{S}(\boldsymbol{X})\|_\infty < \epsilon. \tag{C12}$$

We divide the approximation into two steps by the triangular inequality: 1) approximate $f$ by a Sumformer $\mathcal{S}$ and 2) approximate $\mathcal{S}$ by a Primphormer $\mathcal{T}_{\mathrm{Pri}}$.

$$\sup_{\boldsymbol{X} \in \mathcal{X}^N} \|f(\boldsymbol{X}) - \mathcal{T}_{\mathrm{Pri}}(\boldsymbol{X})\|_\infty \leq \sup_{\boldsymbol{X} \in \mathcal{X}^N} \|f(\boldsymbol{X}) - \mathcal{S}(\boldsymbol{X})\|_\infty + \sup_{\boldsymbol{X} \in \mathcal{X}^N} \|\mathcal{S}(\boldsymbol{X}) - \mathcal{T}_{\mathrm{Pri}}(\boldsymbol{X})\|_\infty. \tag{C13}$$

According to Theorem C.2, we know that there exists a Sumformer $\mathcal{S}$ which approximates $f$ to an error of $\epsilon/2$. This Sumformer has the inherent latent dimension $d'$.

Secondly, we turn to the second term and construct a Primphormer that can approximate Sumformer to $\epsilon/2$ error. The structure of Transformer is $\boldsymbol{X} + \mathrm{FFN}\left(\boldsymbol{X} + \mathrm{Att}(\boldsymbol{X})\right)$ where FFN and Att are the feed-forward and self-attention modules, respectively. The attention map $\mathrm{Att}(\boldsymbol{X})$ of Primphormer is calculated in the primal space (2.8) and the rest of the architecture in Primphormer stays the same. Here, we follow the proof idea proposed in (Alberti et al., 2023) and refer readers to this work for detailed information on the theoretical result.

We have the input $\boldsymbol{X} = [\boldsymbol{x}_1, \cdots, \boldsymbol{x}_N] \in \mathcal{X}^N$ with $\boldsymbol{x}_i \in \mathbb{R}^d$. Set the attention in the first layers to zero, we obtain the feed-forward layers without attention. We first map $\boldsymbol{X}$ with a feed-forward transformation to

$$\begin{bmatrix} \boldsymbol{x}_1 & \cdots & \boldsymbol{x}_N \\ \boldsymbol{x}_1 & \cdots & \boldsymbol{x}_N \end{bmatrix} \in \mathbb{R}^{2d \times N}. \tag{C14}$$

Then, a two-layer feed-forward network can be constructed to act as the identity on the first $N$ components while approximating the function $\boldsymbol{\xi}$ in Sumformer (Hornik et al., 1989; Alberti et al., 2023). We have.

$$\begin{bmatrix} \boldsymbol{x}_1 & \cdots & \boldsymbol{x}_N \\ \boldsymbol{\xi}(\boldsymbol{x}_1) & \cdots & \boldsymbol{\xi}(\boldsymbol{x}_N) \end{bmatrix} \in \mathbb{R}^{(d+d') \times N}. \tag{C15}$$

Before getting to the second step, we add a linear mapping with

$$\begin{cases} \boldsymbol{W} = \begin{bmatrix} \mathbf{0}_{d \times 1} & \boldsymbol{I}_d & \mathbf{0}_{d \times d'} & \mathbf{0}_{d \times d'} \\ \mathbf{0}_{d' \times 1} & \mathbf{0}_{d' \times d} & \boldsymbol{I}_{d'} & \mathbf{0}_{d' \times d'} \end{bmatrix}^{\top} \in \mathbb{R}^{(1+d+2d') \times (d+d')}, \\ \boldsymbol{b} = \begin{bmatrix} \mathbf{1}_N & \mathbf{0}_{N \times (d+2d')} \end{bmatrix}^{\top} \in \mathbb{R}^{(1+d+2d') \times N}, \end{cases} \tag{C16}$$

and get an output after the first step:

$$\begin{bmatrix} 1 & \cdots & 1 \\ \boldsymbol{x}_1 & \cdots & \boldsymbol{x}_N \\ \boldsymbol{\xi}(\boldsymbol{x}_1) & \cdots & \boldsymbol{\xi}(\boldsymbol{x}_N) \\ \mathbf{0}_{d' \times 1} & \cdots & \mathbf{0}_{d' \times 1} \end{bmatrix} \in \mathbb{R}^{(1+d+2d') \times N}. \tag{C17}$$

Secondly, we turn to the attention scheme to represent the sum $\boldsymbol{\Xi} = \sum_{i=1}^{N} \boldsymbol{\xi}(\boldsymbol{x}_i)$ defined in the definition (C.1). Set $\boldsymbol{W}_q = \boldsymbol{W}_k = [\boldsymbol{e}_1, \boldsymbol{0}_{(1+d+2d')\times(d+2d')}]$ with $\boldsymbol{e}_1 = [1, \boldsymbol{0}_{1\times(d+2d')}]^\top$. we have,

$$\phi_q(\boldsymbol{X}_1) = \phi_k(\boldsymbol{X}_1) = \left[\boldsymbol{1}_{N\times 1}, \boldsymbol{0}_{N\times(d+2d')}\right]^\top \in \mathbb{R}^{(1+d+2d')\times N}. \tag{C18}$$

Let the data-dependent projection $f(\boldsymbol{X}) = \boldsymbol{B}\boldsymbol{X}\boldsymbol{1}_N\boldsymbol{1}_{N_s}^\top$ with $\boldsymbol{B} = [\boldsymbol{0}_{d'\times 1}, \boldsymbol{0}_{d'\times d}, \boldsymbol{I}_{d'}, \boldsymbol{0}_{d'\times d'}]$, we have,

$$f(\boldsymbol{X}) = \overbrace{\left[\sum_{i=1}^{N} \boldsymbol{\xi}(\boldsymbol{x}_i), \cdots, \sum_{i=1}^{N} \boldsymbol{\xi}(\boldsymbol{x}_i)\right]}^{N_s} = [\boldsymbol{\Xi}, \cdots, \boldsymbol{\Xi}] \in \mathbb{R}^{d'\times N_s}. \tag{C19}$$

Let $\boldsymbol{W}_e = \boldsymbol{W}_r = [\boldsymbol{e}_1, \boldsymbol{0}_{(1+d+2d')\times(N_s-1)}]^\top$, the projection scores in (2.8) are

$$\begin{cases} \boldsymbol{e}(\boldsymbol{X}_1) = f(\boldsymbol{X}_1)\boldsymbol{W}_e\phi_q(\boldsymbol{X}_1) &= [\boldsymbol{\Xi}, \cdots, \boldsymbol{\Xi}] \in \mathbb{R}^{d'\times N}. \\ \boldsymbol{r}(\boldsymbol{X}_1) = f(\boldsymbol{X}_1)\boldsymbol{W}_r\phi_k(\boldsymbol{X}_1) &= [\boldsymbol{\Xi}, \cdots, \boldsymbol{\Xi}] \in \mathbb{R}^{d'\times N}. \end{cases} \tag{C20}$$

To fit the dimension of the output, we concatenate the projection scores $[\boldsymbol{e}(\boldsymbol{X}_1); \boldsymbol{r}(\boldsymbol{X}_1)] \in \mathbb{R}^{2d'\times N}$, and choose a compatibility matrix $\boldsymbol{W}_c = [\boldsymbol{0}_{(1+d+d')\times 2d'}; \frac{1}{2}\boldsymbol{I}_{d'}, \frac{1}{2}\boldsymbol{I}_{d'}] \in \mathbb{R}^{(1+d+2d')\times 2d'}$, such that

$$\boldsymbol{o}(\boldsymbol{X}_1) = \boldsymbol{W}_c \begin{bmatrix} \boldsymbol{e}(\boldsymbol{X}_1) \\ \boldsymbol{r}(\boldsymbol{X}_1) \end{bmatrix} = \begin{bmatrix} \boldsymbol{0}_{(1+d+d')\times 1} & \cdots & \boldsymbol{0}_{(1+d+d')\times 1} \\ \boldsymbol{\Xi} & \cdots & \boldsymbol{\Xi} \end{bmatrix} \in \mathbb{R}^{(1+d+2d')\times N}. \tag{C21}$$

Then apply a residual connection and obtain the same output as outlined in (Alberti et al., 2023),

$$\begin{bmatrix} 1 & \cdots & 1 \\ \boldsymbol{x}_1 & \cdots & \boldsymbol{x}_N \\ \boldsymbol{\xi}(\boldsymbol{x}_1) & \cdots & \boldsymbol{\xi}(\boldsymbol{x}_N) \\ \boldsymbol{\Xi} & \cdots & \boldsymbol{\Xi} \end{bmatrix} \in \mathbb{R}^{(1+d+2d')\times N}. \tag{C22}$$

Because only the attention map $\mathrm{Att}(\mathrm{X})$ is changed in the architecture and the rest stays the same, the construction of $\psi$ is as same as that in (Alberti et al., 2023), i.e., $\mathcal{O}(N(\frac{1}{\epsilon})^{dN}/N!)$ feed-forward layers for approximating $\psi$ in the discontinuous case and two feed-forward layers for approximating $\psi$ in the continuous case. Above all, we can construct a Primphormer that approximates the Sumformer to $\epsilon/2$ error. $\qquad\square$

### C.5. Proof details of Theorem 3.3

*Proof.* The proof can be done in a similar way as Theorem 3.2. Firstly, let the target function $f(\boldsymbol{X}) := [g(\boldsymbol{x}_1, \{\boldsymbol{x}_2, \cdots, \boldsymbol{x}_N\}), \cdots, g(\boldsymbol{x}_N, \{\boldsymbol{x}_1, \cdots, \boldsymbol{x}_{N-1}\})]$. Since the target function $f$ is continuous, its component functions $f_1, \cdots, f_N$, i.e., $g$, are also continuous. The compactness of $\mathcal{X}$ shows that $\mathcal{X}^N$ is also compact and therefore $g$ is uniformly continuous. Without loss of generality, let the compact support of $g$ be contained in $[0,1]^{d\times N}$. Then we can define a piece-wise constant function $\overline{g}$ by

$$\overline{g}(\boldsymbol{X}) = \sum_{\boldsymbol{P}\in\mathbb{G}_\delta} g(\boldsymbol{P})\boldsymbol{1}\{\boldsymbol{X} \in C_{\boldsymbol{P}}\}, \tag{C23}$$

where the grid $\mathbb{G}_\delta := \{0, \delta, \cdots, 1 - \delta\}^{d\times N}$ for some $\delta := \frac{1}{\Delta}$ with $\Delta \in \mathbb{N}$ consisting of cubes $C_{\boldsymbol{P}} = \prod_{i=1}^{N} \prod_{k=1}^{d} [\boldsymbol{P}_{i,k}, \boldsymbol{P}_{i,k} + \delta)$. Because $g$ is uniformly continuous, for each $\epsilon > 0$, there exists a $\delta > 0$ such that

$$\sup_{\boldsymbol{X}\in\mathcal{X}^N} \|g(\boldsymbol{X}) - \overline{g}(\boldsymbol{X})\|_\infty < \epsilon. \tag{C24}$$

Secondly, choose the positional encoding

$$\boldsymbol{E} = \begin{bmatrix} 0 & 1 & 2 & \cdots & N-1 \\ 0 & 1 & 2 & \cdots & N-1 \\ \vdots & \vdots & \vdots & & \vdots \\ 0 & 1 & 2 & \cdots & N-1 \end{bmatrix} \in \mathbb{R}^{d\times N}. \tag{C25}$$

After applying the quantization, the output is in the following set,

$$\mathbb{H}_\delta := \left\{ \boldsymbol{P} + \boldsymbol{E} \in \mathbb{R}^{d \times N} | \boldsymbol{P} \in \mathbb{G}_\delta \right\}. \tag{C26}$$

Then the $i$-th column of $\boldsymbol{X} + \boldsymbol{E}$ is in the range $[i-1, i)^d$, meaning that the entries corresponding to different tokens lie in disjoint intervals. More precisely, for any $\boldsymbol{H} \in \mathbb{G}_\delta$, its $i$-th column $\boldsymbol{H}_i \in [i - 1 : \delta : i - \delta]$.

Consider a vector $\boldsymbol{u} = \frac{1-\delta}{N\delta^{-d+1}} \times \left( 1, \delta^{-1}, \cdots, \delta^{-d+1} \right) \in \mathbb{R}^d$. It is easy to check that for any $\boldsymbol{H} \in \mathbb{G}_\delta$, the map $l(\boldsymbol{H}_i) = \boldsymbol{u}^\top \boldsymbol{H}_i$ is one-to-one,

$$\boldsymbol{u}^\top \boldsymbol{H}_i \in \left[ \frac{(1-\delta)(i-1)}{N\delta^{-d+1}} \sum_{k=0}^{d-1} \delta^{-k} : \frac{(1-\delta)}{N\delta^{-d}} : \frac{(1-\delta)i}{N\delta^{-d+1}} \sum_{k=0}^{d-1} \delta^{-k} - \frac{(\delta^{-d}-1)}{N\delta^{-d-1}} \right]. \tag{C27}$$

Therefore, for each column $\boldsymbol{H}_i$, the image of $l(\boldsymbol{H}_i)$ is in an interval disjoint from the other columns. We can know that $l(\boldsymbol{H}_i)$ can be thought as a "column id" for different columns, for any permutation $\pi : [N] \to [N]$,

$$l\left( \boldsymbol{H}_{\pi(1)} \right) < l\left( \boldsymbol{H}_{\pi(2)} \right) < \cdots < l\left( \boldsymbol{H}_{\pi(N)} \right). \tag{C28}$$

Besides, it can be easily checked that the image of $l$ lies within the interval $[0, 1]$,

$$0 \le l\left( \boldsymbol{H}_{\pi(1)} \right) < l\left( \boldsymbol{H}_{\pi(2)} \right) < \cdots < l\left( \boldsymbol{H}_{\pi(N)} \right) < 1. \tag{C29}$$

Next, we want to represent $\overline{g}$ using an appropriate $\mathcal{S}$. Without loss of generality, we choose the $k$-th component of $f$, i.e., $\overline{g}(\boldsymbol{x}_k, \{\boldsymbol{x}_i | i \ne k, i \in [N]\})$. Assign each grid point $\boldsymbol{H}$ a coordinate $\chi(\boldsymbol{H}) = \boldsymbol{b} \in [0, 1]^N$ by the construction of the function $l$. Let $\boldsymbol{b} = [l(\boldsymbol{H}_i) | i \in [N]] \in [0, 1]^N$. The map $\chi$ is bijective and there are finitely many $\boldsymbol{b}$. We can enumerate all $\boldsymbol{b}$ using a function $\mu : [0, 1]^N \to \mathbb{N}$. This function could be represented by the Kolmogorov-Arnold representation theorem, as stated below,

**Lemma C.3** (Kolmogorov-Arnold representation (Khesin & Tabachnikov, 2014; Zaheer et al., 2017)). *Let $f : [0, 1]^N \to \mathbb{R}$ be an arbitrary multivariate continuous function iff it has the representation,*

$$f(\boldsymbol{x}_1, \cdots, \boldsymbol{x}_N) = \rho \left( \sum_{n=1}^{N} \lambda_n \phi(\boldsymbol{x}_n) \right) \tag{C30}$$

*with continuous outer and inner functions $\rho : \mathbb{R}^{2N+1} \to \mathbb{R}$ and $\phi : \mathbb{R} \to \mathbb{R}^{2N+1}$. The inner function $\phi$ is independent of the function $f$.*

Now, we can utilize Lemma C.3 to find the representation for the function $\mu$,

$$\mu(\boldsymbol{b}) = \rho \left( \sum_{n=1}^{N} \lambda_n \phi(\boldsymbol{b}_n) \right). \tag{C31}$$

Define $\boldsymbol{\Xi} := \sum_{n=1}^{N} \boldsymbol{\xi}(\boldsymbol{b}_n) = \sum_{n=1}^{N} \lambda_n \phi(\boldsymbol{b}_n)$ and a quantization function $q$ such that $\boldsymbol{b}_n = l(q(\boldsymbol{x}_n + \boldsymbol{E}_n))$. It is feasible because $b_n$ varies for different indices, as claimed in "column id" (C28). Now we can recover the grid $\boldsymbol{H}$,

$$\boldsymbol{H} = \chi^{-1} \circ \mu^{-1} \circ \rho(\boldsymbol{\Xi}). \tag{C32}$$

We then define the function $\psi$ such that the related $\mathcal{S}$ is equal to $\overline{g}$:

$$\psi(\boldsymbol{x}_k, \boldsymbol{\Xi}) := \overline{g} \left( \iota(\chi^{-1} \circ \mu^{-1} \circ \rho(\boldsymbol{\Xi}) - \boldsymbol{E}) \right), \tag{C33}$$

with $\iota : \boldsymbol{P} \mapsto (\boldsymbol{P}_k, \boldsymbol{P}_{i \ne k})$ to fit the input requirement of $\overline{g}$. Since we chose $\overline{g}$ to uniformly approximate $g$, i.e., each component of $f$ up to $\epsilon$ error, it implies that $\mathcal{S}$ with a positional encoding uniformly approximates $f$ up to $\epsilon$ error.

Thirdly, we need to prove the universal approximation between a Sumformer and a Primphormer after adding a positional encoding. The proof (C.4) still holds because it only involves the architecture. We can claim that there exists a Primphormer with a positional encoding $\mathcal{T}_{\mathrm{PE}}$ uniformly approximating a Sumformer $\mathcal{S}$.

Above all, we end the proof by using the triangular inequality,

$$\sup_{\boldsymbol{X} \in \mathcal{X}^N} \|f(\boldsymbol{X}) - \mathcal{T}_{\mathrm{PE}}(\boldsymbol{X})\|_\infty \le \sup_{\boldsymbol{X} \in \mathcal{X}^N} \|f(\boldsymbol{X}) - \mathcal{S}(\boldsymbol{X})\|_\infty + \sup_{\boldsymbol{X} \in \mathcal{X}^N} \|\mathcal{S}(\boldsymbol{X}) - \mathcal{T}_{\mathrm{PE}}(\boldsymbol{X})\|_\infty < \epsilon. \tag{C34}$$

$\square$

## C.6. Expressivity of Primphormer

In this subsection, we prove the expressivity of Primphormer. According to the previous work (Müller & Morris, 2024), which demonstrates that the standard Transformer can simulate the 1-WL test, we prove that Primphormer is also capable of simulating the 1-WL test. This result indicates that the primal representation preserves expressivity, ensuring that Primphormer remains a powerful graph learning model.

**Lemma C.4** (Theorem VIII.4, (Grohe, 2021)). *Let $G = (V, E, \ell)$ be a labeled graph with $N$ nodes, adjacency matrix $\mathbf{A}(G)$, and node feature matrix $\boldsymbol{X}^{(0)} := \boldsymbol{H} \in \mathbb{R}^{d \times N}$ consistent with the label $\ell$. Assume a GNN that for each layer, $t > 0$, updates the node feature matrix,*

$$\boldsymbol{X}^{(t)} := \mathrm{FFN}\left(\boldsymbol{X}^{(t-1)} + 2\boldsymbol{X}^{(t-1)}\mathbf{A}(G)\right). \tag{C35}$$

*Then, for all $t \geq 0$, there exists a parameterization of FFN such that*

$$C_t^1(v) = C_t^1(w) \iff \boldsymbol{X}^{(t)}(v) = \boldsymbol{X}^{(t)}(w), \tag{C36}$$

*for all nodes $v, w \in V$, where $C_t^1$ is the coloring function of the 1-WL test at $t$-th iteration.*

The previous work (Müller & Morris, 2024) has shown that the standard graph Transformer i.e., 1-GT, with sufficiently adjacency-identifying structural embeddings such as LAP/LapPE (Kreuzer et al., 2021) and SPE (Huang et al., 2024) could simulate the (C35), demonstrating the expressive power of the standard graph Transformer is as same as the 1-WL test.

**Lemma C.5** (Theorem 2, (Müller & Morris, 2024)). *Let $G = (V, E, \ell)$ be a labeled graph with $N$ nodes, and node feature matrix $\boldsymbol{X}^{(0)} := \boldsymbol{H} \in \mathbb{R}^{d \times N}$ consistent with the label $\ell$. Then, for all iterations $t \geq 0$, there exists a parameterization of 1-GT such that*

$$C_t^1(v) = C_t^1(w) \iff \boldsymbol{X}^{(t)}(v) = \boldsymbol{X}^{(t)}(w), \tag{C37}$$

*for all nodes $v, w \in V$, where $C_t^1$ is the coloring function of the 1-WL test at $t$-th iteration.*

In our approach, we replace the standard attention module with the primal representation "Prim", forming a model structure $\mathrm{FFN}(\boldsymbol{X} + \mathrm{Prim}(\boldsymbol{X}))$. Our goal is to show that the primal representation preserves the expressivity of the standard Transformers. To ensure a fair comparison, we follow the same setup used by Müller & Morris (2024). Let $G = (V, E, \ell)$ be a labeled graph with $N$ nodes, and node feature matrix $\boldsymbol{X}^{(0)} := \boldsymbol{H} \in \mathbb{R}^{d \times N}$ consistent with the label $\ell$. Then, we initialize $N$ node embedding $\boldsymbol{X}^{(0)} := \boldsymbol{H} + \boldsymbol{P}$, where $\boldsymbol{P} \in \mathbb{R}^{d \times N}$ is the structural embeddings, encoding structural information for each node. For each node $v$,

$$\boldsymbol{P}(v) := \mathrm{FFN}\left(\deg(v) + \mathrm{PE}(v)\right), \tag{C38}$$

where $\deg : V \to \mathbb{R}^l$ is a learnable embedding of the node degree, $\mathrm{PE} : V \to \mathbb{R}^l$ is a node-level PE such as LAP and SPE, and $\mathrm{FFN} : \mathbb{R}^l \to \mathbb{R}^d$ is a feed-forward layer.

**Definition C.6** (LAP and SPE (Müller & Morris, 2024)). Let $(\boldsymbol{\lambda}, \boldsymbol{V})$ denotes the eigensystem of the graph Laplacian $\boldsymbol{L}$ of graph $G$ with $N$ nodes where $\boldsymbol{\lambda} := (\lambda_1, \cdots, \lambda_n)$ is the vector of the $n$ smallest eigenvalues and $\boldsymbol{V} \in \mathbb{R}^{N \times n}$ is the corresponding matrix of eigenvectors. Then the LAP and SPE are defined as follows,

$$\begin{aligned} \mathrm{LAP}(\boldsymbol{\lambda}, \boldsymbol{V}) &= \rho_1\left([\psi(\boldsymbol{V}_1^\top, \lambda), \cdots, \psi(\boldsymbol{V}_N^\top, \lambda)]\right) \\ \mathrm{SPE}(\boldsymbol{\lambda}, \boldsymbol{V}) &= \rho_2\left([\boldsymbol{V}\phi_1(\boldsymbol{\lambda})\boldsymbol{V}^\top, \cdots, \boldsymbol{V}\phi_m(\boldsymbol{\lambda})\boldsymbol{V}^\top]\right), \end{aligned} \tag{C39}$$

where LAP: $\psi : \mathbb{R}^2 \to \mathbb{R}^l$ is a feed-forward layer applied row-wise and $\rho_1 : \mathbb{R}^{N \times l} \to \mathbb{R}^{N \times l}$ is a permutation-equivariant neural network; SPE: $m$ is a hyper-parameter, $\phi_1, \cdots, \phi_m : \mathbb{R}^n \to \mathbb{R}^n$ and $\rho_2 : \mathbb{R}^{N \times N \times m} \to \mathbb{R}^{N \times l}$ are permutation-equivariant neural networks. The graph Laplacian is defined as $\boldsymbol{L} := \boldsymbol{D} - \mathbf{A}$ where $\boldsymbol{D}$ and $\mathbf{A}$ are the diagonal degree matrix and the adjacency matrix of graph $G$, respectively. The normalized graph Laplacian is defined as $\tilde{\boldsymbol{L}} := \boldsymbol{I} - \boldsymbol{D}^{-\frac{1}{2}}\mathbf{A}\boldsymbol{D}^{-\frac{1}{2}}$.

According to Definition C.6, there exists a parameterization of LAP and SPE to represent the original graph Laplacian. This intuition is also proven in Sec. 3.5 in Kreuzer et al. (2021), Sec. 3.4 in Huang et al. (2024), and Theorems. 20 and 22 in Müller & Morris (2024).

**Lemma C.7** (Initialization of input (Müller & Morris, 2024)). *Let $G$ be a graph with node features $\boldsymbol{H} \in \mathbb{R}^{d \times N}$ with the degree function $\deg : V \to \mathbb{N}$. For every tokenization $\boldsymbol{X}^{(0)} = \boldsymbol{H} + \boldsymbol{P}$, there exists a parameterization of $\boldsymbol{X}^{(0)}$ such that for node $v \in V$,*

$$\boldsymbol{X}^{(0)}(v) = \left[\boldsymbol{H}'(v); \mathbf{0}; \deg'(v); \boldsymbol{P}'(v)\right] \in \mathbb{R}^d, \tag{C40}$$

*with $\boldsymbol{P}'(v) = \text{FFN}'\left(\deg'(v) + \text{PE}(v)\right)$, such that $\boldsymbol{H}'(v) \in \mathbb{R}^s, \boldsymbol{0} \in \mathbb{R}^s, \deg'(v) \in \mathbb{R}^r$, and $\text{FFN}' : \mathbb{R}^d \to \mathbb{R}^k$ for $d = 2s + r + k$, and for every $v, w \in V$, it holds,*

$$\begin{aligned}
\boldsymbol{H}(v) = \boldsymbol{H}(w) &\iff \boldsymbol{H}'(v) = \boldsymbol{H}'(w) \\
\deg(v) = \deg(w) &\iff \deg'(v) = \deg'(w) \\
\boldsymbol{P}(v) = \boldsymbol{P}(w) &\iff \boldsymbol{P}'(v) = \boldsymbol{P}'(w).
\end{aligned} \tag{C41}$$

*Moreover, if structure embedding $\boldsymbol{P}$ can recover the graph Laplacian, so does $\boldsymbol{P}'$.*

Using the above Lemma, we can integrate node and structure information to a parameterization of $\boldsymbol{X}^{(0)}$ with a proper dimension while maintaining unchanged common features.

Next, we give the proof of Theorem 3.4 in the main body.

**Theorem C.8.** *Let $G = (V, E, \ell)$ be a labeled graph with $N$ nodes, and node feature matrix $\boldsymbol{X}^{(0)} := \boldsymbol{H} \in \mathbb{R}^{d \times N}$ consistent with the label $\ell$. Then, for all iterations $t \geq 0$, there exists a parameterization of Primphormer such that*

$$C_t^1(v) = C_t^1(w) \iff \boldsymbol{X}^{(t)}(v) = \boldsymbol{X}^{(t)}(w), \tag{C42}$$

*for all nodes $v, w \in V$, where $C_t^1 : V \to \mathbb{N}$ is the coloring function of the 1-WL test at $t$-th iteration.*

*Proof.* According to Lemma C.7, there exists a parameterization of the initialization $\boldsymbol{X}^{(0)}$ such that

$$\boldsymbol{X}^{(0)}(v) = \left[ \boldsymbol{H}'(v); \boldsymbol{0}; \deg'(v); \boldsymbol{P}'(v) \right],$$

for each $v \in V$ and

$$\begin{aligned}
\boldsymbol{H}(v) = \boldsymbol{H}(w) &\iff \boldsymbol{H}'(v) = \boldsymbol{H}'(w) \\
\deg(v) = \deg(w) &\iff \deg'(v) = \deg'(w) \\
\boldsymbol{P}(v) = \boldsymbol{P}(w) &\iff \boldsymbol{P}'(v) = \boldsymbol{P}'(w).
\end{aligned}$$

with $d = 2s + r + k$. We use the induction method to prove it.

First, according to the definition, we have

$$C_0^1(v) = C_0^1(w) \iff \boldsymbol{H}(v) = \boldsymbol{H}(w).$$

Denote $\boldsymbol{H}^{(t)}(v)$ as the representation of the color of node $v$ at iteration $t$. We set $\boldsymbol{H}^{(t)}(v) = \boldsymbol{H}'(v)$, $\boldsymbol{D}_{\text{emb}} \in \mathbb{R}^{r \times N}$ such that for $i$-th column $\boldsymbol{D}_{\text{emb,i}} = \deg'(v_i)$ where $v_i$ is the $i$-th node in a fix but arbitrary node ordering. Then, $\boldsymbol{X}^{(0)}$ can be rewritten as

$$\boldsymbol{X}^{(0)}(v) = \left[ \boldsymbol{H}^{(0)}(v); \boldsymbol{0}; \deg'(v); \boldsymbol{P}'(v) \right],$$

and in matrix form,

$$\boldsymbol{X}^{(0)} = \left[ \boldsymbol{H}^{(0)}; \boldsymbol{0}; \boldsymbol{D}_{\text{emb}}; \boldsymbol{P}' \right] \in \mathbb{R}^{d \times N}.$$

Secondly, suppose the statement holds to iteration $t, t \geq 0$. For the induction, we want,

$$C_{t+1}^1(v) = C_{t+1}^1(w) \iff \boldsymbol{H}^{(t+1)}(v) = \boldsymbol{H}^{(t+1)}(w),$$

which means that the first element of $\boldsymbol{X}^{(t+1)}$ should match the 1-WL-equivalent aggregation,

$$\boldsymbol{X}^{(t+1)} = \left[ \boldsymbol{H}^{(t+1)}; \boldsymbol{0}; \boldsymbol{D}_{\text{emb}}; \boldsymbol{P}' \right] \in \mathbb{R}^{d \times N}. \tag{C43}$$

Recall Lemma C.4, we know the 1-WL-equivalent aggregation follows,

$$\boldsymbol{H}^{(t+1)} := \text{FFN}_{\text{WL}} \left( \boldsymbol{H}^{(t)} + 2\boldsymbol{H}^{(t)}\mathbf{A}(G) \right), \tag{C44}$$

where $\text{FFN}_{\text{WL}}$ is the feed-forward layer to update colors. Then we only need to show that our Primphormer can simulate (C44) to match (C43),

$$\boldsymbol{o}(\boldsymbol{x}) = \boldsymbol{W}_c \begin{bmatrix} \boldsymbol{e}(\boldsymbol{x}) \\ \boldsymbol{r}(\boldsymbol{x}) \end{bmatrix} \quad \text{with} \quad \begin{cases} \boldsymbol{e}(\boldsymbol{x}) = f_X \boldsymbol{W}_e \phi_q(\boldsymbol{x}), \\ \boldsymbol{r}(\boldsymbol{x}) = f_X \boldsymbol{W}_r \phi_k(\boldsymbol{x}), \end{cases} \quad f_X = \boldsymbol{F} + \boldsymbol{B}\boldsymbol{X}\mathbf{1}_N\mathbf{1}_{N_s}^\top. \tag{C45}$$

By setting $\boldsymbol{W}_c = [\boldsymbol{I}, \boldsymbol{0}]$, $N_s = s$, $\boldsymbol{B} = \boldsymbol{0}$, and $\boldsymbol{F} = \boldsymbol{I}$ (the identity matrix), we can parameterize Primphormer as follows:

$$\boldsymbol{o}(\boldsymbol{x}) = \boldsymbol{e}(\boldsymbol{x}) = \boldsymbol{W}_e \boldsymbol{\phi}_q(\boldsymbol{x}), \tag{C46}$$

where we set $\boldsymbol{\phi}_q(\boldsymbol{x}) := \boldsymbol{q}(\boldsymbol{x})/\|\boldsymbol{q}(\boldsymbol{x})\|_2$ and $\boldsymbol{\phi}_k(\boldsymbol{x}) := \boldsymbol{k}(\boldsymbol{x})/\|\boldsymbol{k}(\boldsymbol{x})\|_2$ with $\boldsymbol{q}(x) = \boldsymbol{W}_q \boldsymbol{x}$ and $\boldsymbol{k}(x) = \boldsymbol{W}_k \boldsymbol{x}$. We re-state the projection weight $\boldsymbol{W}_q$ and $\boldsymbol{W}_k$ with expanded sub-matrices to fit $\boldsymbol{X}^{(t)}$ as

$$\boldsymbol{W}_q = \left[\boldsymbol{W}_q^1, \boldsymbol{W}_q^2, \boldsymbol{W}_q^3, \boldsymbol{W}_q^4\right] \in \mathbb{R}^{d \times d},$$
$$\boldsymbol{W}_k = \left[\boldsymbol{W}_k^1, \boldsymbol{W}_k^2, \boldsymbol{W}_k^3, \boldsymbol{W}_k^4\right] \in \mathbb{R}^{d \times d}, \tag{C47}$$

where $\boldsymbol{W}_q^1, \boldsymbol{W}_k^1 \in \mathbb{R}^{d \times s}$, $\boldsymbol{W}_q^2, \boldsymbol{W}_k^2 \in \mathbb{R}^{d \times s}$, $\boldsymbol{W}_q^3, \boldsymbol{W}_k^3 \in \mathbb{R}^{d \times r}$, and $\boldsymbol{W}_q^4, \boldsymbol{W}_k^4 \in \mathbb{R}^{d \times k}$. Then, we define the corresponding output,

$$\boldsymbol{o}(\boldsymbol{X}^{(t)}) := \boldsymbol{W}_e \boldsymbol{\phi}_q(\boldsymbol{X}^{(t)}) = \boldsymbol{W}_e \boldsymbol{q}(\boldsymbol{X}^{(t)}) \mathrm{diag}(\|\boldsymbol{q}(\boldsymbol{X}^{(t)})\|_{2,\mathrm{col}})^{-1} = \boldsymbol{W}_e \boldsymbol{W}_q \boldsymbol{X}^{(t)} \mathrm{diag}(\|\boldsymbol{W}_q \boldsymbol{X}^{(t)}\|_{2,\mathrm{col}})^{-1}, \tag{C48}$$

where $\|\boldsymbol{A}\|_{2,\mathrm{col}} := [\|\boldsymbol{A}_1\|_2, \cdots, \|\boldsymbol{A}_N\|_2]$ denotes the $l_2$-norm of the each column of $\boldsymbol{A}$. According to the KKT conditions (C2), we have $\boldsymbol{W}_e = \sum_{j=1}^{N} \boldsymbol{h}_{r_j} \boldsymbol{\phi}_k(\boldsymbol{x}_j)^{\top}$, in matrix form $\boldsymbol{W}_e = \boldsymbol{H}_{r_j} \boldsymbol{\phi}_k(\boldsymbol{X}^{(t)})^{\top}$, indicating that the row space of $\boldsymbol{W}_e$ is spanned by $\{\boldsymbol{\phi}_k(\boldsymbol{x}_j)^{\top}\}_j$. Thus, we can re-parameterize $\boldsymbol{W}_e$ in the row space such that

$$\boldsymbol{W}_e = \overline{\boldsymbol{H}} \boldsymbol{\phi}_k(\boldsymbol{X}^{(t)})^{\top} = \begin{bmatrix} \boldsymbol{h}_1^1 \cdots & \boldsymbol{h}_N^1 \\ \boldsymbol{h}_1^2 \cdots & \boldsymbol{h}_N^2 \\ \boldsymbol{h}_1^3 \cdots & \boldsymbol{h}_N^3 \\ \boldsymbol{h}_1^4 \cdots & \boldsymbol{h}_N^4 \end{bmatrix} \boldsymbol{\phi}_k(\boldsymbol{X}^{(t)})^{\top} \tag{C49}$$

where $\boldsymbol{h}_i^1 \in \mathbb{R}^s, \boldsymbol{h}_i^2 \in \mathbb{R}^s, \boldsymbol{h}_i^3 \in \mathbb{R}^r, \boldsymbol{h}_i^4 \in \mathbb{R}^k, \forall i \in [N]$ are weight vectors. Now we can formulate the output of our primal representation,

$$\begin{aligned} \boldsymbol{o}(\boldsymbol{X}^{(t)}) &= \overline{\boldsymbol{H}} \boldsymbol{\phi}_k(\boldsymbol{X}^{(t)})^{\top} \boldsymbol{\phi}_q(\boldsymbol{X}^{(t)}) \\ &= \overline{\boldsymbol{H}} \mathrm{diag}(\|\boldsymbol{W}_k \boldsymbol{X}^{(t)}\|_{2,\mathrm{col}})^{-1} \boldsymbol{X}^{(t)^{\top}} \boldsymbol{W}_k^{\top} \boldsymbol{W}_q \boldsymbol{X}^{(t)} \mathrm{diag}(\|\boldsymbol{W}_q \boldsymbol{X}^{(t)}\|_{2,\mathrm{col}})^{-1}. \end{aligned} \tag{C50}$$

By setting $\boldsymbol{W}_q^1, \boldsymbol{W}_q^2, \boldsymbol{W}_q^3, \boldsymbol{W}_k^1, \boldsymbol{W}_k^2, \boldsymbol{W}_k^3$ to zeros, we have

$$\boldsymbol{o}(\boldsymbol{X}^{(t)}) = \overline{\boldsymbol{H}} \mathrm{diag}(\|\boldsymbol{W}_k^4 \boldsymbol{P}'\|_{2,\mathrm{col}})^{-1} \boldsymbol{P}'^{\top} \boldsymbol{W}_k^{4^{\top}} \boldsymbol{W}_q^4 \boldsymbol{P}' \mathrm{diag}(\|\boldsymbol{W}_q^4 \boldsymbol{P}'\|_{2,\mathrm{col}})^{-1}. \tag{C51}$$

Since the LAP and SPE are permutation-invariant functions, they can universally approximate the eigenfunction $f(\boldsymbol{M}) = \boldsymbol{V} \boldsymbol{\lambda}^{\frac{1}{2}} \boldsymbol{M}$ with a permutation matrix $\boldsymbol{M}$ where $\{\boldsymbol{V}, \boldsymbol{\lambda}\}$ is the eigensystem of the (normalized) graph Laplacian. According to Lemma C.7, we know that the structure embedding $\boldsymbol{P}'$ can also recover the (normalized) graph Laplacian, i.e., $\boldsymbol{P}'^{\top} \boldsymbol{P}' = \boldsymbol{L}$. By setting $\boldsymbol{W}_q^4, \boldsymbol{W}_k^4$ as $[\boldsymbol{I}_k; \boldsymbol{0}_{d-k}]$, we have $\boldsymbol{W}_q^4 \boldsymbol{P}' = [\boldsymbol{P}'; \boldsymbol{0}_{d-k}]$ and $\boldsymbol{W}_k^4 \boldsymbol{P}' = [\boldsymbol{P}'; \boldsymbol{0}_{d-k}]$. Then $\boldsymbol{o}$ can be re-parameterized by,

$$\boldsymbol{o}(\boldsymbol{X}^{(t)}) = \overline{\boldsymbol{H}} \mathrm{diag}(\|\boldsymbol{P}'\|_{2,\mathrm{col}})^{-1} \boldsymbol{P}'^{\top} \boldsymbol{P}' \mathrm{diag}(\|\boldsymbol{P}'\|_{2,\mathrm{col}})^{-1}. \tag{C52}$$

Recall the adjacency matrix $\mathbf{A}(G)$ without self-loop and $\boldsymbol{L} = \boldsymbol{D} - \boldsymbol{A}$, we know that $\boldsymbol{L}_{ii} = \boldsymbol{D}_{ii}$, i.e., $\boldsymbol{P}'^{\top}_i \boldsymbol{P}'_i = \boldsymbol{D}_{ii}$, such that $\mathrm{diag}(\|\boldsymbol{P}'\|_{2,\mathrm{col}}) = \boldsymbol{D}^{\frac{1}{2}}$:

$$\boldsymbol{o}(\boldsymbol{X}^{(t)}) = \overline{\boldsymbol{H}} \boldsymbol{D}^{-\frac{1}{2}} \boldsymbol{P}'^{\top} \boldsymbol{P}' \boldsymbol{D}^{-\frac{1}{2}} = \overline{\boldsymbol{H}} \boldsymbol{D}^{-\frac{1}{2}} \boldsymbol{L} \boldsymbol{D}^{-\frac{1}{2}} = \overline{\boldsymbol{H}} \left(\boldsymbol{I} - \boldsymbol{D}^{-\frac{1}{2}} \mathbf{A}(G) \boldsymbol{D}^{-\frac{1}{2}}\right). \tag{C53}$$

By setting $\boldsymbol{h}_i^1, \boldsymbol{h}_i^3$, and $\boldsymbol{h}_i^4$ to zeros and $\boldsymbol{h}_i^2 = \sqrt{|N(i)|} \boldsymbol{H}^{(t)}(i)$, we can obtain the output of the "Prim" module as,

$$\mathrm{Prim}(\boldsymbol{X}^{(t)}) = \boldsymbol{o}(\boldsymbol{X}^{(t)}) = \left[\boldsymbol{0}; \boldsymbol{H}^{(t)} \left(\boldsymbol{D}^{\frac{1}{2}} - \mathbf{A}(G) \boldsymbol{D}^{-\frac{1}{2}}\right); \boldsymbol{0}; \boldsymbol{0}\right]. \tag{C54}$$

Finally, recalling the model structure $\mathrm{FFN}(\boldsymbol{X} + \mathrm{Prim}(\boldsymbol{X}))$, Primphormer computes the next representation $\boldsymbol{X}^{(t+1)}$ as follows:

$$\begin{aligned} \boldsymbol{X}^{(t+1)} &= \mathrm{FFN}\left(\boldsymbol{X}^{(t)} + \mathrm{Prim}(\boldsymbol{X}^{(t)})\right) \\ &= \mathrm{FFN}\left(\left[\boldsymbol{H}^{(t)}; \boldsymbol{0}; \boldsymbol{D}_{\mathrm{emb}}; \boldsymbol{P}'\right] + \left[\boldsymbol{0}; \boldsymbol{H}^{(t)} \left(\boldsymbol{D}^{\frac{1}{2}} - \mathbf{A}(G) \boldsymbol{D}^{-\frac{1}{2}}\right); \boldsymbol{0}; \boldsymbol{0}\right]\right) \\ &= \mathrm{FFN}\left(\left[\boldsymbol{H}^{(t)}; \boldsymbol{H}^{(t)} \left(\boldsymbol{D}^{\frac{1}{2}} - \mathbf{A}(G) \boldsymbol{D}^{-\frac{1}{2}}\right); \boldsymbol{D}_{\mathrm{emb}}; \boldsymbol{P}'\right]\right), \end{aligned} \tag{C55}$$

and for each node $v \in V$,

$$\boldsymbol{X}^{(t+1)}(v) = \text{FFN}\left(\left[\boldsymbol{H}^{(t)}(v); |N(v)|^{\frac{1}{2}}\boldsymbol{H}^{(t)}(v) - \sum_{w \in N(v)} |N(v)|^{-\frac{1}{2}}\boldsymbol{H}^{(t)}(w); \deg'(v); \boldsymbol{P}'(v)\right]\right). \quad \text{(C56)}$$

We can define $\deg'(v) := [|N(v)|^{\frac{1}{2}}; \boldsymbol{0}_{r-1}]$. The domain is obviously compact, thus there exists choices of $f_{\text{FFN}}, f_{\text{lin}_2}, f_{\text{lin}_1}, f_{\deg} : \mathbb{R}^d \to \mathbb{R}^d$ is continuous. We can use a feed-forward layer FFN to approximate $f_{\text{FFN}} \circ f_{\text{lin}_2} \circ f_{\text{lin}_1} \circ f_{\deg}$ arbitrarily close. We define $f_{\text{FFN}}, f_{\text{lin}_2}, f_{\text{lin}_1}, f_{\deg}$ as follows,

$$f_{\deg}\left(\left[\boldsymbol{H}^{(t)}(v); |N(v)|^{\frac{1}{2}}\boldsymbol{H}^{(t)}(v) - \sum_{w \in N(v)} |N(v)|^{-\frac{1}{2}}\boldsymbol{H}^{(t)}(w); \deg'(v); \boldsymbol{P}'(v)\right]\right)$$
$$= \left[\underbrace{\boldsymbol{H}^{(t)}(v)}_{(a)}; \underbrace{\boldsymbol{H}^{(t)}(v) - \sum_{w \in N(v)} |N(v)|^{-1}\boldsymbol{H}^{(t)}(w)}_{(b)}; \deg'(v); \boldsymbol{P}'(v)\right], \quad \text{(C57)}$$

where $f_{\deg}$ multiplies the degree $|N(v)|^{-\frac{1}{2}}$ in to the second component. Next, $f_{\text{lin}_1}$ conducts a linear transformation such that $(b) = 2 \times ((b) - (a))$,

$$f_{\text{lin}_1}\left(\left[\boldsymbol{H}^{(t)}(v); \boldsymbol{H}^{(t)}(v) - \sum_{w \in N(v)} |N(v)|^{-1}\boldsymbol{H}^{(t)}(w); \deg'(v); \boldsymbol{P}'(v)\right]\right)$$
$$= \left[\underbrace{\boldsymbol{H}^{(t)}(v)}_{(a)}; \underbrace{-2\sum_{w \in N(v)} |N(v)|^{-1}\boldsymbol{H}^{(t)}(w)}_{(b)}; \deg'(v); \boldsymbol{P}'(v)\right]. \quad \text{(C58)}$$

Next, $f_{\text{lin}_2}$ conducts a linear transformation such that $(a) = (a) - |N(v)|(b)$ and $(b) = \boldsymbol{0}$,

$$f_{\text{lin}_2}\left(\left[\boldsymbol{H}^{(t)}(v); -2\sum_{w \in N(v)} |N(v)|^{-1}\boldsymbol{H}^{(t)}(w); \deg'(v); \boldsymbol{P}'(v)\right]\right)$$
$$= \left[\underbrace{\boldsymbol{H}^{(t)}(v) + 2\sum_{w \in N(v)} \boldsymbol{H}^{(t)}(w)}_{(a)}; \boldsymbol{0}; \deg'(v); \boldsymbol{P}'(v)\right]. \quad \text{(C59)}$$

Finally, $f_{\text{FFN}}$ updates the first component in the same way as $\text{FFN}_{\text{WL}}$ does in (C44),

$$f_{\text{FFN}}\left(\left[\boldsymbol{H}^{(t)}(v) + 2\sum_{w \in N(v)} \boldsymbol{H}^{(t)}(w); \boldsymbol{0}; \deg'(v); \boldsymbol{P}'(v)\right]\right)$$
$$= \left[\text{FFN}_{\text{WL}}\left(\boldsymbol{H}^{(t)}(v) + 2\sum_{w \in N(v)} \boldsymbol{H}^{(t)}(w)\right); \boldsymbol{0}; \deg'(v); \boldsymbol{P}'(v)\right] \quad \text{(C60)}$$
$$= \left[\boldsymbol{H}^{(t+1)}(v); \boldsymbol{0}; \deg'(v); \boldsymbol{P}'(v)\right].$$

In matrix form,

$$\boldsymbol{X}^{(t+1)} = f_{\text{FFN}} \circ f_{\text{lin}_2} \circ f_{\text{lin}_1} \circ f_{\deg}\left(\left[\boldsymbol{H}^{(t)}; \boldsymbol{H}^{(t)}\left(\boldsymbol{D}^{\frac{1}{2}} - \mathbf{A}(G)\boldsymbol{D}^{-\frac{1}{2}}\right); \boldsymbol{D}_{\text{emb}}; \boldsymbol{P}'\right]\right)$$
$$= \left[\boldsymbol{H}^{(t+1)}; \boldsymbol{0}; \boldsymbol{D}_{\text{emb}}; \boldsymbol{P}'\right], \quad \text{(C61)}$$

which completes the proof. $\square$

According to Lemma C.5 and Theorem C.8, we know that both Transformer and Primphormer can simulate the 1-WL test in terms of distinguishing non-isomorphic graphs.

# D. Pseudo-code

**Algorithm 1** PyTorch-like Pseudo-Code for Primphormer Module.

```python
import torch
import torch.nn as nn
import torch.nn.functional as F
from torch_geometric.nn import global_mean_pool
from torch_geometric.utils import to_dense_batch

class Primphormer(nn.Module):
    def __init__(self, in_dim, out_dim, n_heads, Ns, low_rank):
        super().__init__()
        self.d_keys = out_dim // n_heads # key dimension.
        self.q_proj = nn.Linear(in_dim, out_dim) # query
        self.k_proj = nn.Linear(in_dim, out_dim) # key
        self.vn_proj = nn.Linear(in_dim, out_dim) # virtual node
        self.n_heads = n_heads

        self.We = nn.Parameter(nn.init.orthogonal_(torch.Tensor(Ns, n_heads, self.d_keys)))
        self.Wr = nn.Parameter(nn.init.orthogonal_(torch.Tensor(Ns, n_heads, self.d_keys)))
        self.Lambda = nn.Parameter(nn.init.uniform_(torch.Tensor(n_heads, low_rank)))
        self.concate_weight = nn.Linear(2*low_rank, self.d_keys)

    def feature_map(self, Q, K):
        Q = F.normalized(Q, p=2, dim=-1)
        K = F.normalized(K, p=2, dim=-1)
        return Q, K

    def propagate_vn(self, batch, h):
        h = self.vn_proj(h)
        h_vn = global_mean_pool(h, batch.batch).unsqueeze(1) # aggregate by the virtual node.
        fx = h_vn + batch.fx # update f_X by the virtual node.
        return fx

    def forward(self, batch):
        x = batch.x
        x_dense, mask = to_dense_batch(x, batch.batch)
        B, M = mask.shape # batch, maximal #nodes
        fx = self.propagate_vn(batch, x)
        Q = self.q_proj(x_dense).view(B, M, self.n_heads, -1)
        K = self.k_proj(x_dense).view(B, M, self.n_heads, -1)
        Q, K = self.feature_map(Q, K)

        # compute data-dependent projections
        We_X = torch.einsum('bdv,vhe->bdhe', fx.transpose(2, 1), self.We)
        Wr_X = torch.einsum('bdv,vhe->bdhe', fx.transpose(2, 1), self.Wr)

        # compute projection scores
        escore = torch.einsum('bmhd,bhde->bmhe', Q, We_X.permute(0, 2, 3, 1))[mask]
        rscore = torch.einsum('bmhd,bhde->bmhe', K, Wr_X.permute(0, 2, 3, 1))[mask]

        score = torch.cat((escore, rscore), dim=-1)
        out = self.concate_weight(score).contiguous()
        out = out.view(-1, self.n_heads * self.d_keys) # final output
        batch.fx = fx #update for the next layer

        loss_escore = (torch.einsum('nhd,hd->nhd', escore, self.Lambda).norm(dim=-1,p=2)**2).mean() / 2
        loss_rscore = (torch.einsum('nhd,hd->nhd', rscore, self.Lambda).norm(dim=-1,p=2)**2).mean() / 2
        loss_trace = torch.einsum('dhe,ehk->dhk', self.We.permute(2, 1, 0), self.Wr).mean(dim=1).trace()
        loss_svd = (loss_escore + loss_rscore - loss_trace) ** 2

        return out, loss_svd
```

---

**Algorithm 2** Algorithm for Primphormer in the GPS architecture.

---

**Input:** Graph $G = (V, E)$ with $N$ nodes and $M$ edges; Adjacency matrix $\boldsymbol{A} \in \mathbb{R}^{N \times N}$; Node features $\boldsymbol{X} \in \mathbb{R}^{d_{\mathrm{n}} \times N}$, Edge features $\boldsymbol{E} \in \mathbb{R}^{d_{\mathrm{e}} \times M}$; Node and edge encoders; Local message passing model instance $\mathtt{MPNN}_e$; Primphormer model instance $\mathtt{Prim}$; Positional encoding function $f_{\mathrm{PE}}$; Layers $l \in [L-1]$.

**Output:** Node representations $\boldsymbol{X}^L \in \mathbb{R}^{d \times N}$ and edge representations $\boldsymbol{E}^L \in \mathbb{R}^{d \times M}$ for downstream tasks.

1: $\boldsymbol{P}_{\mathrm{node}}, \boldsymbol{P}_{\mathrm{edge}} \leftarrow \varnothing$;
2: $\boldsymbol{P}_{\mathrm{node}}, \boldsymbol{P}_{\mathrm{edge}} \leftarrow f_{\mathrm{PE}}(\boldsymbol{X}, \boldsymbol{E})$
3: $\boldsymbol{X}^1 \leftarrow \bigoplus_{\mathrm{node}} (\mathtt{NodeEncoder}(\boldsymbol{X}), \boldsymbol{P}_{\mathrm{node}})$
4: $\boldsymbol{E}^1 \leftarrow \bigoplus_{\mathrm{edge}} (\mathtt{EdgeEncoder}(\boldsymbol{E}), \boldsymbol{P}_{\mathrm{edge}})$
5: **for** $l = 1, \cdots, L-1$ **do**
6: $\quad \hat{\boldsymbol{X}}_M^{l+1}, \boldsymbol{E}^{l+1} \leftarrow \mathtt{MPNN}_e^l (\boldsymbol{X}^l, \boldsymbol{E}^l, \boldsymbol{A})$
7: $\quad \hat{\boldsymbol{X}}_P^{l+1} \leftarrow \mathtt{Prim}^l (\boldsymbol{X}^l)$
8: $\quad \boldsymbol{X}_M^{l+1} \leftarrow \mathtt{BatchNorm} \left( \mathtt{Dropout} \left( \hat{\boldsymbol{X}}_M^{l+1} \right) + \boldsymbol{X}^l \right)$
9: $\quad \boldsymbol{X}_P^{l+1} \leftarrow \mathtt{BatchNorm} \left( \mathtt{Dropout} \left( \hat{\boldsymbol{X}}_P^{l+1} \right) + \boldsymbol{X}^l \right)$
10: $\quad \boldsymbol{X}^{l+1} \leftarrow \mathtt{MLP}^l \left( \boldsymbol{X}_M^{l+1} + \boldsymbol{X}_P^{l+1} \right)$
11: **end for**
12: **Return:** $\boldsymbol{X}^L$ and $\boldsymbol{E}^L$

---

# E. Additional experiments

We also conduct experiments to compare against more models (Ma et al., 2023). We report the experimental results in Table A5.

Table A5 Comparisons between our method and GRIT(Ma et al., 2023).

| MODEL | CIFAR10 | | | MNIST | | |
|---|---|---|---|---|---|---|
| GPS | ACC↑ | TIME(S/EPOCH) | MEMORY(GB) | ACC↑ | TIME(S/EPOCH) | MEMORY(GB) |
| PRIMPHORMER | $74.13 \pm 0.241$ | 32.6 | 2.74 | $98.56 \pm 0.042$ | 43.7 | 1.71 |
| GRIT(MA ET AL., 2023) | $76.46 \pm 0.881$ | 158.8 | 22.8 | $98.11 \pm 0.111$ | 70.1 | 7.69 |

We report the performance drop of removing $f_X$ in the following table,

Table A6 The removal impact performance of $f_X$.

| | PASCALVOC | COCO | PEPTIDES-FUNC | PEPTIDES-STRUCT | PCQM |
|---|---|---|---|---|---|
| PRIMPHORMER | $0.4602 \pm 0.0077$ | $0.3903 \pm 0.0061$ | $0.6612 \pm 0.0065$ | $0.2495 \pm 0.0008$ | $0.3757 \pm 0.0079$ |
| NO $f_X$ | $0.4513 \pm 0.0089$ | $0.3758 \pm 0.0082$ | $0.6509 \pm 0.0072$ | $0.2576 \pm 0.0011$ | $0.3516 \pm 0.0126$ |

