# OpenReview forum: "Primphormer: Efficient Graph Transformers with Primal Representations"
_ICML.cc/2025/Conference — ICML 2025 poster_

### Official Review · Reviewer_3RwF · 2025-03-10

**Overall Recommendation:** 3

**Summary:**

This paper proposes a novel Graph Transformer model, named Primphormer, which models the self-attention mechanism in the primal space, avoiding costly pair-wise computations and enabling an efficient variant of Graph Transformers. By introducing an additional primal objective loss, Primphormer achieves high efficiency in terms of both runtime and memory usage, allowing for larger and deeper neural networks and enabling larger batch sizes, thereby enhancing the model's capacity and generalization ability. Furthermore, Primphormer preserves expressive power equivalent to that of traditional Transformers, effectively distinguishing non-isomorphic graphs. Experimental results on various graph benchmarks demonstrate the effectiveness and efficiency of the proposed Primphormer.

**Claims And Evidence:**

The paper provides convincing experimental results and theoretical evidence to support its claims.

**Essential References Not Discussed:**

The paper overlooks Graph ViT [1] and GRIT [2] from ICML 2023, as well as GEAET [3] from ICML 2024, which could serve as meaningful comparison baselines.

[1] X He, et al. A generalization of vit/mlp-mixer to graphs. ICML, 2023.

[2] L Ma, et al. Graph inductive biases in transformers without message passing. ICML, 2023.

[3] J Liang, M Chen and J Liang. Graph External Attention Enhanced Transformer. ICML, 2024.

**Experimental Designs Or Analyses:**

I have reviewed the experimental design of this paper and found some issues. Specifically, the paper does not compare the latest baselines. For instance, it does not include comparisons with the GRIT [2] model and Graph ViT [1] model from ICML 2023, nor the GEAET [3] model from ICML 2024. These models have significant impact in the field of graph representation learning, and the lack of comparison could affect the comprehensive evaluation of Primphormer's performance.

[1] X He, et al. A generalization of vit/mlp-mixer to graphs. ICML, 2023.

[2] L Ma, et al. Graph inductive biases in transformers without message passing. ICML, 2023.

[3] J Liang, M Chen and J Liang. Graph External Attention Enhanced Transformer. ICML, 2024.

**Methods And Evaluation Criteria:**

The proposed methods and evaluation criteria are suitable for the given problem and application context.

**Other Comments Or Suggestions:**

None.

**Other Strengths And Weaknesses:**

Strengths:
-The proposed method is innovative.
-The theoretical analysis is thorough and detailed.

Weaknesses：
-The performance of the proposed method is not particularly outstanding, and several models from 2023/2024 are not compared, such as GRIT, Graph ViT from ICML 2023, and GEAET from ICML 2024.
-The paper claims to be an Efficient Graph Transformer, but it lacks testing on large graph datasets and only evaluates on graph-level tasks.

**Questions For Authors:**

1. The proposed method in this paper seems similar to linear attention methods. Could the authors explain the differences between the two approaches? Additionally, could the authors highlight the advantages of the proposed method compared to Graph Transformer methods with linear attention, such as Nodeformer from NeurIPS 2022 and SGFormer from NeurIPS 2023?
2. Could the authors provide performance results on large graph datasets?

**Relation To Broader Scientific Literature:**

The key contribution of the paper is the introduction of Primphormer, based on primal space, which improves efficiency by avoiding pairwise computations. Unlike existing methods, Primphormer uses a dual approach to reduce computational complexity while maintaining expressiveness.

**Theoretical Claims:**

I have reviewed the theoretical content of this paper, and there are no significant issues. For example, Theorem 2.2's duality derivation and the introduction of the KKT conditions are sound and consistent with related optimization theory. Lemma 2.3 effectively shows that the solutions in the dual space lead to a zero-valued objective in the primal space, which aligns with common results in duality theory. Furthermore, Theorem 3.2 provides a rigorous proof of the approximation ability of Primphormer for any continuous function, confirming its strong performance. Overall, the theoretical derivations in the paper are correct.

---

> ### Author Rebuttal · Authors · 2025-03-31
>
> Thank you for your appreciation of the innovation of Primphormer and insightful comments. We address your concerns below:
>
> - R4.1: Broader experiments.
>
> > Thank you for mentioning the latest baselines, Graph ViT/MLP-mixer [1], GRIT [2], and GEAET [3]. Following your suggestion, we have included comparisons with these methods, as shown in the tables below.
>
> > Table 1 Comparison on the CIFAR10 dataset.
> > |  | Acc$\uparrow$| Time(s/epoch)$\downarrow$ | Memory(GB) $\downarrow$|
> > | :--- | :--- | :---: | :---: |
> > |Graph MLP-Mixer|  73.96±0.33    |   47.5   | 3.11 |
> > |Graph ViT      |  72.11±0.55    |   49.7   | 3.05 |
> > |GRIT           |$\small{\textbf{76.46}}$±$\small\textbf{0.88}$|158.8 | 22.8 |
> > |GEAET          |  76.33±0.43    | 45.8     | 4.9 |
> > |Primphormer    |  74.13±0.24    | $\small{\textbf{32.6}}$ | $\small{\textbf{2.7}}$ |
>
> > Table 2 Comparison on the MNIST dataset.
> > |  | Acc$\uparrow$| Time(s/epoch)$\downarrow$ | Memory(GB) $\downarrow$|
> > | :--- | :--- | :---: | :---: |
> > |Graph MLP-Mixer|97.42±0.11|57.2|2.26|
> > |Graph ViT      |97.25±0.23|53.6|2.25|
> > |GRIT           |98.11±0.11|70.1|7.69|
> > |GEAET          |98.41±0.09|49.2|2.11|
> > |Primphormer    |$\small{\textbf{98.56}}$±$\small{\textbf{0.04}}$|$\small{\textbf{43.7}}$|$\small{\textbf{1.71}}$|
> >
> > It can be observed that our method achieves the lowest time and memory costs while maintaining competitive performance, which aligns with the motivation of this paper.
>
> > We have also conducted experiments on large graph datasets for node-level tasks, including ogbn-arxiv, ogbn-proteins, and Amazon2m.
> >
> >|  | ogbn-arxiv |  ogbn-proteins  |   Amazon2m   |
> >|:--------|:--------|:-------|:------|
> >| \#Nodes  |  169,343 |  132,534  | 2,449,029 |
> >| \#Edges  | 1,166,243| 39,561,252| 61,589,140 |
> >|NodeFormer |59.90±0.42|77.45±1.15| 87.85±0.24 |
> >|SGFormer   |72.63±0.13|$\small{\textbf{79.53}}$±$\small{\textbf{0.38}}$| 89.09±0.10|
> >|Primphormer|$\small{\textbf{73.10}}$±$\small{\textbf{0.24}}$|78.93±0.31|$\small{\textbf{90.33}}$±$\small{\textbf{0.32}}$|
> >
> > We hope these results contribute to a comprehensive evaluation of Primphormer's performance.
>
> - R4.2: Discussion with other linear attention methods.
>
> > Linear attention mechanisms, such as NodeFormer [4] and SGFormer [5], aim to reduce computational complexity by decomposing or approximating the kernel matrix, operating in the dual space. For example, NodeFormer uses a random feature-based approach, while SGFormer drops the softmax activation to approximate the kernel matrix. In contrast, our method adopts a technically different approach by leveraging the asymmetric kernel trick. Instead of operating in the dual space, we directly model the representation of attention outputs in the primal space.
>
> We will include the experiments and discussion in the final version of the manuscript.
>
> [1] X He, et al. A generalization of vit/mlp-mixer to graphs. ICML, 2023.
>
> [2] L Ma, et al. Graph inductive biases in transformers without message passing. ICML, 2023.
>
> [3] J Liang, M Chen and J Liang. Graph External Attention Enhanced Transformer. ICML, 2024.
>
> [4] Wu Q, et al. Nodeformer: A scalable graph structure learning transformer for node classification. NeurIPS, 2022.
>
> [5] Wu Q, et al. SGFormer: Simplifying and empowering transformers for large-graph representations[J]. NeurIPS, 2023.

---

> > ### Comment · Reviewer_3RwF · 2025-04-02
> >
> > Thank you for your response. I have carefully reviewed your response and intend to increase my rating to 2.

---

> > > ### Author Response · Authors · 2025-04-03
> > >
> > > We sincerely appreciate your constructive feedback and the elevated score.
> > >
> > > Your suggestions regarding experiments with the latest baselines and large graphs are very insightful. As a method to improve efficiency, the reported results and additional experiments you mentioned support the improvement of efficiency, which have greatly enhanced our paper.
> > >
> > > Our method takes a different approach to reducing the cost of the self-attention mechanism by leveraging the primal-dual relationship. Unlike other methods, the proposed approach approximates the output of the self-attention rather than the attention scores. A discussion comparing this method with other linear attention methods will also help readers better understand our contributions.

---

### Official Review · Reviewer_1bgR · 2025-03-11

**Overall Recommendation:** 3

**Summary:**

This paper introduces an efficient graph transformer, called Primphormer,  that addresses the quadratic complexity issue of traditional graph transformers by using a primal representation. The authors showed that Primphormer serves as a universal approximator for functions on both sequences and graphs, retaining its expressive power for distinguishing non-isomorphic graphs. Experiments on various graph benchmarks demonstrate that Primphormer achieves competitive empirical results.

## update after rebuttal
By considering the additional experiments, I keep my positive score.

**Claims And Evidence:**

1. Reduced computational complexity: The paper provides analysis showing that the primal representation used in Primphormer has linear complexity ($O(Nps$) time and $O(2N_ss + 2Np)$ memory), which is significantly more efficient than the quadratic complexity ($O(N^2s$) time and $O(N^2 + Ns)$ memory) of traditional graph transformers.
2. Theoretical properties: The paper demonstrates through Theorems 3.2 and 3.3 that Primphormer can approximate any continuous function on sequences and graphs arbitrarily well.

**Essential References Not Discussed:**

Please see the Experimental design and analysis, some efficient graph transformers should be cited and compared.

**Experimental Designs Or Analyses:**

Primphormer is tested on a wide range of graph benchmarks, and compared against multiple relevant baselines.The experimental design also includes both running time and memory usage comparisons, which is crucial for validating their claim of improved computational efficiency.  One concern is that it may need to be compared with some more efficient graph transformer approaches, such as [1], [2].

[1] Polynormer: Polynomial-Expressive Graph Transformer in Linear Time, ICLR 2024
[2] SGFormer: Simplifying and Empowering Transformers for Large-Graph Representations, NeurIPS 2023

**Methods And Evaluation Criteria:**

This paper evaluates Primphormer on diverse benchmark datasets, including LRGB, standard GNN benchmarks, molecular datasets, large-scale graph, and graph isomorphism benchmark (BREC). These datasets cover a broad range of graph types and tasks, making them appropriate for comprehensive evaluation.

**Other Comments Or Suggestions:**

1. Maybe it is better to organize the experimental results according to different tasks, e.g., for node-level, link-level, and graph-level. Now they are mixed.
2. What is the KSVD on Line 770?

**Other Strengths And Weaknesses:**

Strengths:
1. The paper introduces a novel primal representation for graph transformers that addresses the quadratic complexity issue inherent in traditional self-attention mechanisms.
2. Primphormer is proved to be as powerful as Transformer in terms of distinguishing non-isomorphic graphs.
3. Primphormer is tested on a wide range of graph benchmarks and shows competitive performance.

Weaknesses:
1. The main contribution of this paper is to design efficient graph transformer, reducing the quadratic complexity to linear. However, it lacks comparisons with other complexity reduction approaches, e.g., some recent advances in efficient attention mechanisms, to name a few [1], [2].
2. The presentation of this paper could be improved to make it easier to follow, e.g., in the theoretical results section, explain how these theorems relate to the main claims of this paper.

[1] Polynormer: Polynomial-Expressive Graph Transformer in Linear Time, ICLR 2024
[2] SGFormer: Simplifying and Empowering Transformers for Large-Graph Representations, NeurIPS 2023

**Questions For Authors:**

I'd like to see more discussions/evaluations with other efficient graph transformer approaches.

**Relation To Broader Scientific Literature:**

This paper proposes a new appraoch of efficient graph transformers.  The proposed Primphformer builds on established concepts from kernel machines, and universal approximation theory, while specifically addressing the unique challenges posed by graph data.

**Theoretical Claims:**

I checked the proofs in Appendix C.

---

> ### Author Rebuttal · Authors · 2025-03-31
>
> Thank you for your appreciation of the novelty of Primphormer. We address your concerns below:
>
> - R3.1: Broader experiments.
>
> > Following your suggestion, we have compared our method with other efficient graph Transformers such as Polynormer and SGFormer, as illustrated in the following table.
> >| Acc$\uparrow$ | Computer |  Photo  |   CS   | Physics | WikiCS |
> >|:--------|:--------|:-------|:------|:-------|:------- |
> >| \#Nodes  |  13,752  |  7,650  | 18,333 | 34,493  | 11,701 |
> >| \#Edges  | 245,861  | 119,081 | 81,894 | 247,962 | 216,123 |
> >|SGFormer   |92.32±1.66|94.28±1.36| 95.21±1.14 | 96.65±1.26 | 79.48±0.96 |
> >|Exphormer  |91.59±0.31|95.27±0.42| 95.03±0.09 | $\small{\textbf{97.16}}$±$\small\textbf{0.48}$ | 78.54±0.49 |
> >|Polynormer |$\small{\textbf{93.38}}$±$\small\textbf{0.13}$|  96.01±0.12  |95.50±0.18|97.12±0.12|79.64±0.67|
> >|Primphormer|92.47±0.55|$\small{\textbf{96.22}}$±$\small\textbf{0.29}$|$\small{\textbf{95.66}}$±$\small\textbf{0.22}$|97.02±0.17|$\small{\textbf{80.11}}$±$\small\textbf{0.84}$|
> >
> > We observed that Primphormer outperforms on three of the five datasets. We hope these results contribute to a comprehensive evaluation of Primphormer's performance.
>
> - R3.2: Presentation, organization, and KSVD.
>
> > Thank you for your insightful suggestions and careful reading. Following your suggestions, we will continue to improve the presentation, particularly in the theoretical section. We believe your feedback will enhance the readability of the manuscript.
>
> > In this paper, we followed a commonly used experimental organization approach in the graph learning community [1,2]. As we are unable to update the PDF during the author-reviewer discussion period, we will revisit the organization based on your suggestion and select one approach for the final version of the manuscript.
>
> > Thank you for your detailed comment. KSVD stands for Kernel Singular Value Decomposition. The KSVD optimization problem mentioned on Line 770 corresponds to our primal optimization problem outlined in Equation 2.5. We will revise this part in the manuscript to improve clarity.
>
> We will include the experiments and discussion in the final version of the manuscript.
>
> [1] Shirzad H, et al. Exphormer: Sparse transformers for graphs. ICML, 2023.
>
> [2] L Ma, et al. Graph inductive biases in transformers without message passing. ICML, 2023.

---

> > ### Comment · Reviewer_1bgR · 2025-04-04
> >
> > Thank the authors for the additional experiments, and I'll keep my positive rating, though it is another paper addressing the scalability in graph transformers.

---

> > > ### Author Response · Authors · 2025-04-04
> > >
> > > Thank you for maintaining your positive rating. We sincerely appreciate your recognition of our work, and your valuable suggestions have greatly contributed to improving it.

---

### Official Review · Reviewer_3vhH · 2025-03-16

**Overall Recommendation:** 4

**Summary:**

This work presents Primphormer, a graph transformer architecture that leverages a primal-dual framework to reformulate the self-attention mechanism for graphs, which has previously been done for self attention for sequences. Unlike previous graph transformers (such as GraphGPS with a global vanilla Transformer) that compute pairwise attention, leading to quadratic complexity, Primphormer derives a primal representation by integrating a virtual node that aggregates global graph information with learned projection weights. This design reduces the computational and memory costs as expected but also retains the expressiveness and shows competitive performance with prior baselines of efficient graph transformers.

**Claims And Evidence:**

- The paper claims that Primphormer can universally approximate continuous functions on graphs while significantly reducing runtime complexity compared to standard GTs.
- This is supported by- theoretical analysis demonstrating that the primal representation derived via an asymmetric kernel trick which avoids quadratic pairwise computations by integrating global information through a virtual node.
- Empirical results across a diverse set of benchmarks (e.g., CIFAR10, MalNet-Tiny, ogbn-products) that show competitive performance with improved memory and runtime efficiency over baselines such as Exphormer and traditional Transformer-based approaches.
- While the use of virtual nodes through f_x transformation is essential as claimed, how does its removal impact performance drop?
- In addition, a single virtual node may introduce a bottleneck in aggregating global information because of being a single aggregation point. How can this be resolved in principle and keep the overall Primphormer benefits?
- Also, the evidence for efficiency gains are not clear in Table 4, and the corresponding claim does not seem justified fully in section 4, page 6 (except when graph size is high as in malnet or ogbn-products).

**Essential References Not Discussed:**

NA

**Experimental Designs Or Analyses:**

Experiment design is done by replacing the global attention component in GraphGPS framework with proposed method, which makes it a fair design.

**Methods And Evaluation Criteria:**

The method is implemented by having feature maps for queries and keys with learned projection weights (W_e and W_r) and a virtual node via a data-dependent aggregation function (f_X) that captures global graph info and ensures permutation equivariance.

Evaluation is done by replacing the global attention component in GraphGPS with proposed method, and on public graph benchmarks that were used in GraphGPS and subsequent efficient graph transformers.

**Other Comments Or Suggestions:**

NA

**Other Strengths And Weaknesses:**

In terms of understanding the contributions of the paper, there may be a concerns, the efficiency gains are not seen clearly in the experiments and the overall contribution is to extend primal attention from general sequential input to graph input while addressing the challenges during generalization.

**Questions For Authors:**

already included in above sections, eg. in Claims And Evidence section.

**Relation To Broader Scientific Literature:**

The paper is good positioned within the current literature on scalable/efficient graph transformers and corresponding primal attention works in general transformer literature (eg. Chen et al, 2023).

**Theoretical Claims:**

A key theoretical claim is that Primphormer is a universal approximator for functions on both sequences and graphs despite using a sparse, efficient attention mechanism. This is justified using the tools employed in previous methods.

---

> ### Author Rebuttal · Authors · 2025-03-31
>
> Thank you for your insightful suggestions. We address your concerns below:
>
> - R2.1: The removal impact performance of $f_X$.
>
> > Following your insightful suggestion, we report the performance drop of removing $f_X$ in the following table.
> >|     |  PascalVOC   |     COCO     | Peptides-Func | Peptides-Struct |    PCQM      |
> >|:------:|:------------:|:------------:|:-------------:|:---------------:|:-------------:|
> >| Metric | F1$\uparrow$ | F1$\uparrow$ | AP$\uparrow$  | MAE$\downarrow$ | MRR$\uparrow$ |
> >| Primphormer | 0.4602 ± 0.0077| 0.3903 ± 0.0061 | 0.6612 ± 0.0065   | 0.2495 ± 0.0008  | 0.3757 ± 0.0079 |
> >| No $f_X$ | 0.4513 ± 0.0089| 0.3758 ± 0.0082 | 0.6509 ± 0.0072   | 0.2576 ± 0.0011  | 0.3516 ± 0.0126 |
> >
> > Removing $f_X$ results in lower performance and a higher standard deviation, highlighting its importance in contributing to the model's stability and effectiveness.
>
> - R2.2: Virtual nodes.
>
> > To address the bottleneck in aggregating global information, we can explore the use of multiple or hierarchical virtual nodes:
>
> > (1) Multiple virtual nodes: Introducing multiple virtual nodes instead of a single one can help distribute the burden of global aggregation. Each virtual node could focus on aggregating information from a subset of nodes within the graph. These subsets could be obtained by applying graph partitioning methods [1] before training, thereby reducing the bottleneck effect.
> >
> > (2) Hierarchical virtual nodes: Employing hierarchical virtual nodes enables a multi-level aggregation process. For example, virtual nodes at lower layers could aggregate local information, while higher-level virtual nodes could focus on capturing global information [2].
>
> >Overall, your comment is highly insightful, and we consider this topic a valuable direction for future research.
>
> - R2.3: Clearer efficiency gains.
>
> > Thank you for your question regarding the gains. Indeed, the previous table may not have been very clear, as it included three metrics across several tasks. To better illustrate the improvements, we now directly report the differences rather than the absolute values. Additionally, we provide the average performance across all tasks for better clarity. The relative gain is defined as $R:=\frac{\delta}{L}$, where $\delta$ represents the difference between the compared method and GPS+Transformer, and $L$ is the value of GPS+Transformer.
>
> >  We present the triplet $(R_A, R_T, R_M)$ where $R_A, R_T$, and $R_M$ denote the relative gains in accuracy, running time, and peak memory usage, respectively. The desired outcome is to achieve a higher $R_A\uparrow$, alongside lower $R_T\downarrow$ and $R_M\downarrow$, reflecting reduced running time and memory usage.
> >| $(R_A\uparrow, R_T\downarrow, R_M\downarrow)$ | Cifar10 | MalNet. |PascalVOC | Peptides-Func | Average |
> >| :--- | :--- | :--- | :--- | :--- | :--- |
> >| GPS+BigBird    | (-2.53\%, 0.971, -0.262)  | (-1.24\%, 0.401, -0.923)    | (-26.07\%, 0.469, -0.362)|(-10.42\%, 3.054, -0.410) | (-10.07\%, 1.224, -0.489) |
> >| GPS+Performer  | (-2.26\%, 0.814, 1.756)   | (-0.92\%, -0.684, -0.672)    | (-0.32\%, 0.396, -0.215)|(-0.92\%, 0.695, -0.089) | (-1.11\%, 0.305, 0.195) |
> >| GPS+Exphormer  | (3.29\%, 0.589, 0.454)   | (0.56\%, -0.732, -0.706)    | (6.40\%, -0.011, -0.060)|(-0.12\%, -0.406, -0.431)| (2.53\%, -0.140, -0.186) |
> >| GPS+Prim-Atten |(-1.02\%, 0.146, -0.281)  | (-0.57\%, -0.731, -0.927)   | (-16.38\%, -0.278, -0.394)|(-1.35\%, -0.383, -0.601) | (-4.83\%, -0.311, $\small\textbf{-0.550}$) |
> >| GPS+Primphormer| (2.52\%, 0.164, -0.281)  | (0.13\%, -0.733, -0.919)   | (6.53\%, -0.289, -0.396)|(1.18\%, -0.398, -0.597) | ($\small\textbf{2.59}$\%, $\small\textbf{-0.314}$, -0.548) |
> >
> > The table above shows that Primphormer achieves the highest relative gains in accuracy and running time on average, as well as the second-highest (and very close to the best) gain in memory usage. We hope this helps to demonstrate the efficiency gains more clearly.
>
> We will include the discussion and additional experiments in the final version of the manuscript.
>
> [1] Çatalyürek Ü, et al. More recent advances in (hyper) graph partitioning. ACM Computing Surveys, 2023.
>
> [2] Vonessen C, et al. Next Level Message-Passing with Hierarchical Support Graphs. ArXiv:2406.15852, 2024.

---

> > ### Comment · Reviewer_3vhH · 2025-04-05
> >
> > I thank the authors for their time in clarifying my questions on virtual nodes, its bottlenecks and the efficiency gains, which overall shows the benefits of the final Primphormer architecture. I increase the score thereby.

---

> > > ### Author Response · Authors · 2025-04-05
> > >
> > > Thank you for taking the time to carefully review our manuscript and for recognizing the benefits of the final Primphormer architecture. We truly appreciate your thoughtful feedback and the increased score, which motivates us to continue refining our work.

---

### Official Review · Reviewer_kDDD · 2025-03-16

**Overall Recommendation:** 4

**Summary:**

The paper aims to bypass the scale-restricting quadratic complexity of graph transformers with a primal representation of self-attention. This is accomplished by extending the linear-complexity primal representation of self-attention on sequences presented in [1]. The authors identify the lack of ordering in graph data (necessary for permutation equivariance) and the lack of flexibility of the data-adaptive weight as issues in [1]. These are addressed in optimization problem (2.5) with a virtual node allowing for global information collection under permutation equivariance which is then incorporated into projection matrices enabling a data driven dual basis. The duality of (2.5) is established under KKT conditions yielding a primal representation in which pairwise computation can be avoided relying on an asymmetric kernel trick.

The paper continues with an analysis of Primphormer’s universal approximation property on sequences and graphs, followed by its expressivity in regards to the 1-Weisfeiler-Leman test for subgraph isomorphism. Experiments follow to demonstrate Primphormer’s performance, efficiency, and expressivity compared to baselines for a variety of common datasets and their corresponding tasks.


[1] Chen, Y., Tao, Q., Tonin, F., and Suykens, J. A. K. Primal-attention: Self-attention through asymmetric kernel svd in primal representation. In Thirty-seventh Conference on Neural Information Processing Systems, 2023.

**Claims And Evidence:**

Yes

**Essential References Not Discussed:**

No

**Experimental Designs Or Analyses:**

Did you check the soundness/validity of any experimental designs or analyses? Please specify which ones, and discuss any issues.

Overall the experimental designs and analysis are fine. However, Primphormer and GraphGPS+Transformer Table 1 and Table 3 the results for PascalVOC-SP do not match. I was not sure if this was simply an oversight. Otherwise, I was curious as to why Primphormer would perform consistently better than GraphGPS+Transformer given the analysis on performance and expressivity.

Originally, I thought it odd that Table 5 did not include 1-WL given Corollary 3.5. However, if I remember correctly GraphGPS+Transformer and Graphormer, etc. with position information are more powerful than 1-WL, while at least Graphormer is upper-bounded by 3-WL per [2]. I believe this is the reason for its exclusion. I still would be interested in seeing the results of 1-WL just for the sake of comparison.

[2] Jiarui Feng, Yixin Chen, Fuhai Li, Anindya Sarkar, and Muhan Zhang. 2022. How powerful are K-hop message passing graph neural networks. In Proceedings of the 36th International Conference on Neural Information Processing Systems (NIPS '22). Curran Associates Inc., Red Hook, NY, USA, Article 345, 4776–4790.

**Methods And Evaluation Criteria:**

Yes, Primphormer is compared against several related architectures for a variety of benchmark datasets and their corresponding tasks with the convolutional layer remaining fixed, varying only the self-attention mechanism.

**Other Comments Or Suggestions:**

Nothing notable

**Other Strengths And Weaknesses:**

The other weaknesses of the paper are addressed in other relevant sections. My only additional concern is with the overall novelty as it is primarily an extension of [1] to graphs, but with a data adaptive basis rather than data adaptive weight. However, I find the theoretical and empirical analysis to be quite convincing and help to offset this.

**Questions For Authors:**

Tables 1, 2, and 3 appear to report average and standard deviation but it is not specified in the text for how many runs of each architecture. I was unable to find this from a cursory examination of your supplemental materials. Is it the same average over 5 runs as in Table 5?

**Relation To Broader Scientific Literature:**

The paper advances the state of the art of graph transformers with linear complexity self-attention without the requirement of sparseness in the graph. Though the main concept of a primal-dual formulation is primarily an extension of a previous work on sequence data to graph data, it does have some novel improvement. The analysis and experiments are convincing and Primphormer is something that I would find personally useful.

**Theoretical Claims:**

I found no notable issues with the theoretical claims. However, I was curious as to why Subformer was used for the proof of Theorem 3.2.


I believe that Corollary 3.5 is minorly incorrect and also should have a stronger lower-bound. From the proof of Theorem 3.4 it should require positional/structure features and also that the Transformer with such features is strictly more powerful than 1-WL.

---

> ### Author Rebuttal · Authors · 2025-03-31
>
> Thanks for your insightful comments and appreciation of this work. We address your concerns below:
>
> - R1.1: The reason for using Sumformer.
>
> > We use Sumformer as a bridge to analyze the approximation. Specifically, we decompose the approximation into two parts: (1) Primphormer to Sumformer and (2) Sumformer to the target function. Leveraging Sumformer allows us to avoid the need for exponentially many attention layers, requiring only one attention layer to represent the Sumformer.
>
> - R1.2: Revised Corollary 3.5.
>
> > Thank you for your insightful comment. We agree with your suggestion that Corollary 3.5 should be revised. Since both the Transformer and Primphormer are capable of simulating the 1-WL test, we claim that there exist parameterizations of the Transformer and Primphormer such that the node features (or colors) produced by these models are identical.
>
> > (Revised corollary 3.5.) Let $G=(V,E,\ell)$ with $N$ nodes and feature matrix $X^{(0)}\in\mathbb{R}^{d\times N}$ consistent with the label $\ell$. Then for all iterations $t\geq 0$, there exist parameterizations of Transformer and Primphormer and a positional encoding such that $X^{(t)}\_\mathcal{T}(v)=X^{(t)}\_\mathcal{T}(u)\iff X^{(t)}\_{\rm Pri}(v)=X^{(t)}\_{\rm Pri}(u)$ for all nodes $v,u\in V$ where $X^{(t)}\_\mathcal{T}$ and $X^{(t)}\_{\rm Pri}$ are node features of Transformer and Primphormer, respectively.
>
> - R1.3: Results in Tables 1 and 3.
>
> > Thanks for your careful reading. Table 1 is correct, but the results of Primphormer and GraphGPS+Transformer in Table 3 were not updated (Fortunately, the rankings of the results remain unchanged.). We sincerely apologize for this oversight and ensure that this will be corrected in the final version of the manuscript.
>
> - R1.4: Results of the 1-WL test.
>
> > Table 5 presents the expressive results on the BREC benchmark [1]. The key challenge of this benchmark lies in its distinguishing difficulty, as it includes graphs that are indistinguishable by the (1-WL to 4-WL) tests. Consequently, the 1-WL test fails to distinguish graphs in the BREC benchmark, as demonstrated in the following table.
> >
> >| Model| Basic | Regular |Extension | CFI |
> >| :---: | :---: | :---: | :---: | :---: |
> >| 1-WL  | 0     | 0     | 0     |0      |
>
> - R1.5: The experimental setting regarding runs.
>
> > In our experiments (except Table 5), we reported the average and standard deviation over 10 runs. We will include this description in the final version of the manuscript.
>
> [1] Wang Y, Zhang M. An Empirical Study of Realized GNN Expressiveness. ICML, 2024.

---

### Decision · Program_Chairs · 2025-05-01

**Decision:**

Accept (poster)

**Comment:**

This paper introduces a linear-time self-attention for graphs based on primal representation. All reviewers acknowledged that both the technical and empirical contributions of the paper are solid. The main concerns raised by the reviewers include: (1) missing analysis on the impact of virtual node, (2) concerns regarding information bottleneck in aggregating global graph information into a single virtual node, (3) missing comparisons to efficient graph transformer methods, (4) missing comparisons to graph representation learning approaches, and (5)  missing results on larger graph datasets. The authors provided a comprehensive rebuttal addressing most of these concerns, leading the consensus among reviewers to recommend the acceptance. After reading the paper, reviews, and rebuttal, AC agrees with the reviewers' decision and recommends acceptance.